# Proteostasis in dendritic cells is controlled by the PERK signaling axis independently of ATF4

Andreia Mendes[1,2,3,*] , Julien P Gigan[1,*], Christian Rodriguez Rodrigues[1] , Sébastien A Choteau[1,6] , Doriane Sanseau[4], Daniela Barros[1,2,3], Catarina Almeida[2,3] , Voahirana Camosseto[1,3,4], Lionel Chasson[1], Adrienne W Paton[5], James C Paton[5], Rafael J Argüello[1,4] , Ana-Maria Lennon-Duménil[4], Evelina Gatti[1,2,3,4] , Philippe Pierre[1,2,3,4]

In stressed cells, phosphorylation of eukaryotic initiation factor 2α (eIF2α) controls transcriptome-wide changes in mRNA translation and gene expression known as the integrated stress response. We show here that DCs are characterized by high eIF2α phosphorylation, mostly caused by the activation of the ER kinase PERK (EIF2AK3). Despite high p-eIF2α levels, DCs display active protein synthesis and no signs of a chronic integrated stress response. This biochemical specificity prevents translation arrest and expression of the transcription factor ATF4 during ER-stress induction by the subtilase cytotoxin (SubAB). PERK inactivation, increases globally protein synthesis levels and regulates IFN-β expression, while impairing LPS-stimulated DC migration. Although the loss of PERK activity does not impact DC development, the cross talk existing between actin cytoskeleton dynamics; PERK and eIF2α phosphorylation is likely important to adapt DC homeostasis to the variations imposed by the immune contexts.

## Introduction

DCs are key regulators of both protective immune responses and tolerance to self-antigens (Dalod et al, 2014). DCs are professional APCs, equipped with pattern recognition receptors (PRRs), capable of recognizing microbe-associated molecular patterns (MAMPs) (Akira et al, 2006) and enhance their immunostimulatory activity (Steinman, 2007). MAMPs detection by DCs triggers the process of maturation/activation, which culminates in the unique capacity of priming naïve T cells in lymphoid organs. LPS detection by TLR4 promotes DCs maturation by triggering a series of signaling cascades resulting in secretion of polarizing and inflammatory cytokines, up-regulation of co-stimulatory molecules, as well as

enhanced antigen processing and presentation (Mellman, 2013). All these functions are accompanied by major remodeling of membrane trafficking and actin organization to favor both antigen capture and migration to the lymph nodes (West et al, 2004; Chabaud et al, 2015; Arguello et al, 2016; Bretou et al, 2017).

Upon activation by MAMPs, like LPS, a large augmentation of protein synthesis, representing a two to fivefold increase above resting state, occurs in DCs. This is required for the up-regulation of co-stimulatory molecules at the cell surface and acquires T-cell immune-stimulatory function (Lelouard et al, 2007; Reverendo et al, 2019). The phosphorylation of eukaryotic initiation factor 2 (eIF2) is a central hub for regulating protein synthesis during stress. In homeostatic conditions, eIF2 mediates the assembly of the mRNA translation initiation complex and regulates start codon recognition. During stress, phosphorylation of the α subunit of eIF2 (eIF2α) on serine 51 is mediated by a group of four eIF2α kinases (EIF2AK1-4), which specifically senses physiological imbalance (Arguello et al, 2016; Costa-Mattioli & Walter, 2020). Phosphorylation of eIF2α converts eIF2 into an inhibitor of the GDP–GTP guanidine exchange factor eIF2B, impairing the GDP–GTP recycling required to form new translation initiation complexes (Yamasaki & Anderson, 2008). Consequently, increased eIF2α phosphorylation impacts cells in two main ways: (i) By reducing the rate of translation initiation and thus global protein synthesis levels; (ii) By favoring the translation of the activating transcription factor 4 (ATF4) (Han et al, 2013; Fusakio et al, 2016) which in turn activates the transcription of genes involved in the integrated stress response (ISR) (Costa-Mattioli & Walter, 2020).

The ISR protects cells from amino acid deprivation, oxidative, mitochondrial stress or viral infections, and is also incorporated as a branch of the ER unfolded protein response (UPR) upon PERK-activation. The ISR comprises a negative feedback loop that causes eIF2α dephosphorylation, through the induction of GADD34 (also known as PPP1R15a), a phosphatase 1 (PP1c) co-factor (Novoa et al, 2001; Harding et al, 2009). Dephosphorylation of p-eIF2α by GADD34/PP1c complexes, and

[1]Aix Marseille Université, Centre National de la Recherch Scientifique (CNRS), Institut National de la Santé et de la Recherche Médicale (INSERM), Centre d'Immunologie de Marseille Luminy (CIML), CENTURI, Marseille, France   [2]Department of Medical Sciences, Institute for Research in Biomedicine (iBiMED) and Ilidio Pinho Foundation, University of Aveiro, Aveiro, Portugal   [3]International Associated Laboratory (LIA) CNRS "Mistra", Marseille, France   [4]INSERM U932, Institut Curie, ANR-10-IDEX-0001-02 PSL* and ANR-11-LABX-0043, Paris, France   [5]Department of Molecular and Biomedical Science, Research Centre for Infectious Diseases, University of Adelaide, Adelaide, Australia   [6]Aix-Marseille Université, INSERM, Theories and Approaches of Genomic Complexity (TAGC), CENTURI, Marseille, France

Correspondence: pierre@ciml.univ-mrs.fr; gatti@ciml.univ-mrs.fr
*Andreia Mendes and Julien P Gigan contributed equally to this work

associated protein synthesis restoration, signal ISR termination, and return to cellular homeostasis (Novoa et al, 2001). If stress persists, long-term ATF4 expression promotes programmed cell death, through the induction of the pro-apoptotic transcription factor CHOP (Marciniak et al, 2004). ATF4 also regulates the expression of Rho GTPases and can control cell motility (Pasini et al, 2016), whereas globular actin is part of the PP1c/GADD34 complex and provides additional targeting specificity for dephosphorylating p-eIF2α (Chambers et al, 2015; Chen et al, 2015).

The ISR can enter in a cross talk with specialized MAMPs sensing pathways, which turns on or amplifies inflammatory cytokines production in different cell types including DCs (Claudio et al, 2013; Reverendo et al, 2018). TLR activation in macrophages undergoing an ISR suppress CHOP induction and protein synthesis inhibition, preventing apoptosis in activated cells (Woo et al, 2009). Moreover, ATF4 binds interferon regulatory factor-7 (IRF7) and prevents type-I IFN transcription (Liang et al, 2011). Several key innate immunity signaling cascades are also believed to be dependent for their signalosome assembly on the chaperone HSPB8 and the eIF2α kinase heme-regulated inhibitor (HRI/EIF2AK1) (Pierre, 2019). Microbe-activated HRI was shown to mediate phosphorylation of eIF2α and increase ATF4-dependent expression of HSPB8, thus amplifying signal transduction and inflammatory cytokines transcription in macrophages (Abdel-Nour et al, 2019).

We show here that DCs from spleen or derived from Fms-related tyrosine kinase 3 ligand (Flt3-L) treated-BM cultures display high levels of phosphorylated eIF2α. Using Cre/*lox* recombination to generate mice specifically lacking GADD34 (PPP1R15a) or PERK (EIF2AK3) activity in DCs, we demonstrate that PERK-dependent eIF2α phosphorylation is acquired during BMDC differentiation in vitro. PERK drives high eIF2α phosphorylation in steady-state DCs with a low impact on protein synthesis levels. We found that mRNA translation in DCs, differently to what has been shown during chronic ISR (cISR) (Guan et al, 2017), is mediated despite high p-eIF2α levels by an eIF4F-dependent mechanism. These features endow DC with increased resistance to acute ER stress, preventing ATF4 induction in response to stressors such as the bacterial subtilase cytotoxin (SubAB). We also found that LPS-activated primary DCs rely on PERK and eIF2α phosphorylation to amplify type-I IFN expression, but, conversely to macrophages, not to promote pro-inflammatory cytokines transcription nor IL-1β secretion (Abdel-Nour et al, 2019; Chiritoiu et al, 2019). GADD34 antagonizes PERK activity to maintain functional protein synthesis levels in non-activated DCs and upon stimulation with LPS, contributing directly to DC function by modulating IFN-β expression. PERK activity impacts positively DC migration speed, correlating with the regulation of p-eIF2α levels by the synergistic action of GADD34 and actin cytoskeleton reorganization. Thus, DCs require PERK and GADD34 activity to coordinate protein synthesis, activation, type-I IFN production and migration capacity in response to MAMPs and adapt their biochemical functions to the variations encountered in their external environment.

# Results

### Steady-state DCs display high levels of eIF2α phosphorylation

Physiological levels of phosphorylated eIF2α (p-eIF2α) were monitored in mouse spleen sections by immunohistochemistry. All CD11c+ DC subsets expressing either CD8α (cDC1), CD11b (cDC2), or B220 (plasmacytoïd DC, pDC), displayed high levels of eIF2α phosphorylation (Fig 1A and B), strongly contrasting with other splenocytes, such as B cells (Fig 1B lower panel). Splenocytes isolation and flow cytometry based-quantification of eIF2α phosphorylation confirmed that DC subsets display higher levels of p-eIF2α than T (CD3+/CD4+ or /CD8+) or B cells (Fig 1C). We next evaluated p-eIF2α in BM-derived DCs differentiated in presence of Flt3-Ligand (Flt3-L BMDC), encompassing the major cDC1, cDC2, and pDC subsets in different proportions (circa 30%, 60%, and 10%, respectively) with phenotypes equivalent to those of spleen DC subsets (Brasel et al, 2000). Cell sorting and analysis of the different populations by immunoblot confirmed that all DC subsets display higher eIF2α phosphorylation in comparison with isolated primary CD8[+] T cells or MEFs stimulated or not with the ER-stress inducing drug thapsigargin for 2 h (Fig 1D). Quantification of p-eIF2α/eIF2α ratios indicated that steady-state DCs display two to four times more p-eIF2α, than stressed MEFs, with the cDC1 population displaying the highest ratio of phosphorylation (Fig 1D). We next evaluated when eIF2α phosphorylation was acquired during DC differentiation in vitro. Daily analysis of differentiating Flt3-L BMDCs established that high p-eIF2α levels appear from 4 d of culture (Fig 1E), confirming that eIF2α phosphorylation is an integral part of Flt3-L induced DC differentiation.

Given the dominant negative effect of p-eIF2α on translation initiation, we monitored protein synthesis in the different DC subsets. CD8[+] T cells were used as a reference because in these cells, p-eIF2α is barely detectable. We used puromycilation and detection by flow cytometry (flow) to measure protein synthesis level in splenocytes populations (Schmidt et al, 2009; Arguello et al, 2018). Despite higher eIF2α phosphorylation levels in all resting DC subsets, mRNA translation is five to eight times higher than in resting CD8[+] T cells and close to the levels reached by these cells upon CD3/CD28 stimulation (Fig 1F). We next monitored protein synthesis every 2 d of culture to establish precisely the influence of eIF2α phosphorylation during DC differentiation in vitro. We applied flow cytometry and dimensionality reduction using t-distributed stochastic neighbor embedding (tSNE) to visualize DC differentiation and protein synthesis activity within the different subpopulations over time (Fig 2A). We confirmed that protein synthesis levels steadily increased with the appearance of all three DC subsets, this despite high eIF2α phosphorylation. Noteworthy, the cDC1 population that displays the most elevated level of eIF2α phosphorylation is the DC subset endowed with the highest level of protein synthesis. These observations suggest that steady-state DCs have adapted their translation machinery to overcome the dominant negative effect on translation initiation of eIF2α phosphorylation on Ser51, which is associated with the acquisition of the DC phenotype.

### eIF2B and eIF2A expression is up-regulated upon DC differentiation

P-eIF2α inhibits translation initiation by forming a stable inhibitory complex that reduces the guanidine exchange factor activity of eIF2B. eIF2B is an enzymatic complex with a γ2ε2 sub-units core with levels generally lower than those of its substrate eIF2. Thus, a

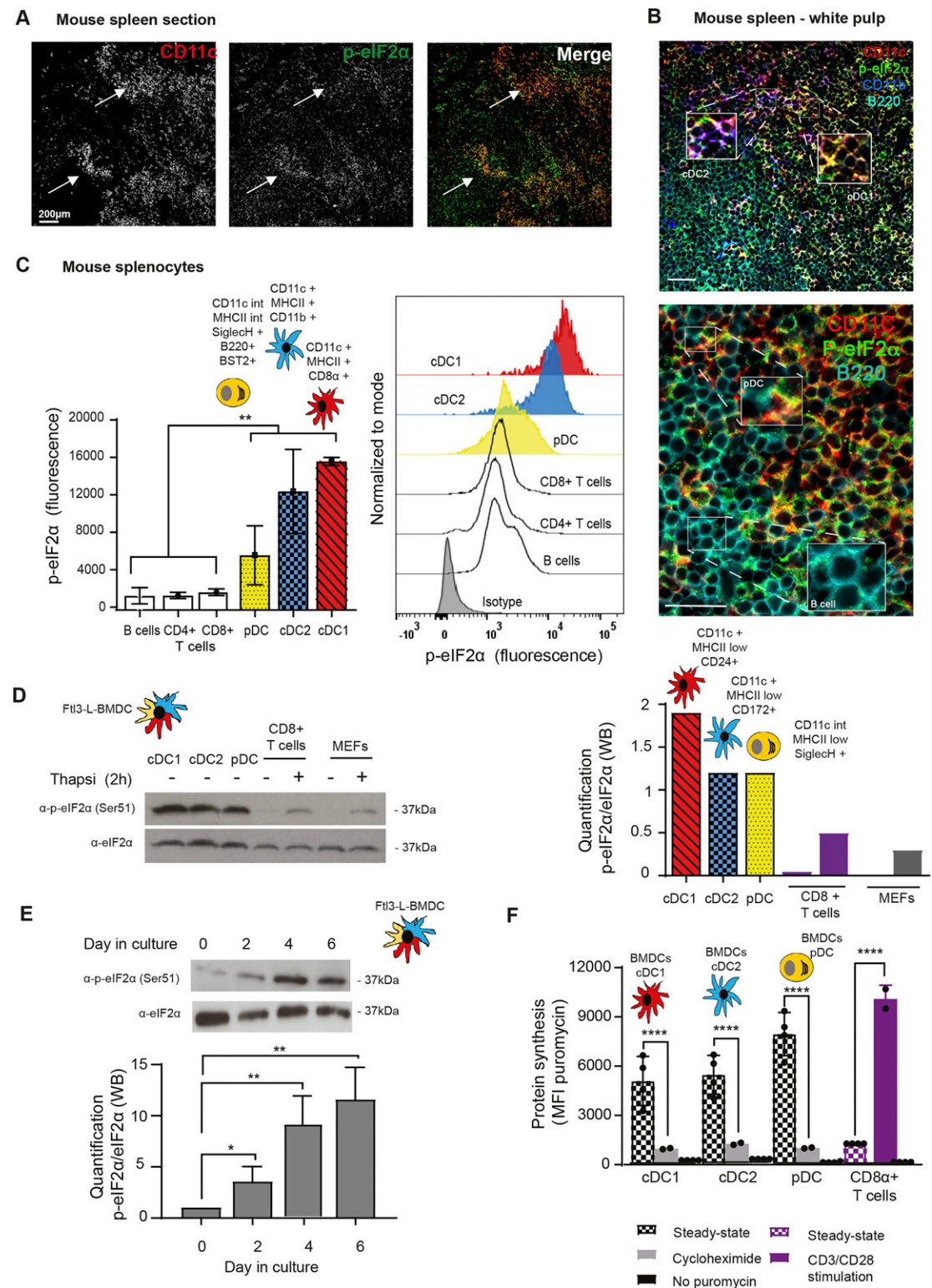

**Figure 1. Steady-state Flt3-L BMDCs and splenic DCs display remarkably high levels of eIF2α without inhibition of translation.**
**(A)** Immunohistochemistry of mouse spleen with staining for CD11c (red) and p-eIF2α (green). Scale bar: 200 μm, magnification: 10×. Single color images are shown and merged picture (right row), high level of p-eIF2α staining is mostly found co-localizing in cells positive for CD11c+ (DCs, white arrowheads). **(B)** Immunohistochemistry of mouse spleen in the white pulp for CD11c (red), p-eIF2α (green), CD11b (blue), and B220 (turquoise). Scale bars: 50 μm, magnification: 40×. In the upper panel, magnified areas show p-eIF2 detection in cDC2 (CD11c+/CD11b+) and cDC1 (CD11c+/CD11b−). In the lower panel, magnified areas show p-eIF2 detection in pDCs (B220+/CD11c+) and in B cells (B220+ and CD11c−). **(C)** Relative p-eIF2α levels measured by flow in different mouse spleen populations. Statistical analysis was performed by Mann–Whitney

partial eIF2α phosphorylation is sufficient to attenuate protein synthesis initiation in most cells (Adomavicius et al, 2019). We monitored, during Flt3-L BMDC differentiation, the expression of the different eIF2B components. From day 2 in culture, all eIF2B subunits levels were increased transcriptionally, and for eIF2Bε, translationally as well (Fig 2B and C). A similar observation was done with eIF2A, a factor involved, in place of eIF2, in the translation of specialized cellular or viral mRNAs (Kim et al, 2011; Starck et al, 2016). The progression of the ratio eIF2Bε, and potentially of eIF2A, over eIF2α expression and phosphorylation during differentiation (Fig 2C) could therefore explain the progressive acquisition by BMDCs of significant protein synthesis levels despite abundant eIF2α phosphorylation.

### Flt3-L BMDCs activation by LPS promotes eIF2α and eEF2 dephosphorylation

Complementary to p-eIF2α, phosphorylated translation elongation factor 2 (eEF2) is a major repressor of translation in adverse growth conditions, such as starvation, or accumulation of misfolded proteins in the ER (Ryazanov, 2002; Lazarus et al, 2017). Like for p-eIF2α (Fig 3A), the high levels of p-eEF2 present in steady-state DCs (Arguello et al, 2018) were gradually decreased during *Escherichia coli* LPS stimulation of TLR4-expressing cDC2 (Fig 3B). eEF2 phosphorylation therefore parallels what is observed for eIF2α in cDC1 and cDC2 (Fig 3A) and could be involved in the control of protein synthesis levels upon DC activation. Given the rapidity and intensity of eIF2α and eEF2 dephosphorylation upon activation, we applied to cDC2 the SunRISE technique, a method for monitoring translation elongation intensity using flow (Arguello et al, 2018). cDC2 displayed a striking augmentation of translation intensity upon LPS activation compared with the steady-state situation (T = 0 s), quasi doubling its level in 6 h (Fig 3C and D). Polysomes elongation speed, indicated by the rate of puromycin staining decay after harringtonine treatment (slope), was also increased (x2) by LPS (Fig 3D). eIF2α and eEF2 dephosphorylation are correlated with increased mRNA translation initiation and elongation allowing protein synthesis to reach its maximum concomitantly to the acquisition by DC of their full immune-stimulatory capacities (Lelouard et al, 2007).

### PPP1R15a (GADD34) controls eIF2α dephosphorylation in activated DCs

The inducible PP1c co-factor PPP1R15a, known as GADD34, is key in mediating p-eIF2α dephosphorylation in the resolution phase of the ISR during the UPR (Novoa et al, 2001, 2003). Interestingly, GADD34 induction was reported in inflammatory situations or upon

MAMPs stimulation of different immune cell subsets (Clavarino et al, 2012b, 2016; Ito et al, 2015). In MEF, GADD34 expression is necessary for the production of IFN-β upon concomitant sensing of cytosolic dsRNA by RIG I-like-helicases and activation of protein kinase RNA-activated (PKR)-dependent phosphorylation of eIF2α (Clavarino et al, 2012a).

To further explore the importance of GADD34 in the control eIF2α pathway in DC, we generated a novel transgenic mouse model with floxed alleles for *Ppp1r15a/Gadd34*. This modification in the *Ppp1r15a* gene allows, upon Cre recombinase expression, the deletion of the third exon that codes for the C-terminal PP1 interacting domain of GADD34. This deletion creates a null phenotype for GADD34-dependent eIF2α dephosphorylation (Harding et al, 2009) (Fig S1A). *Ppp1r15a*^loxp/loxp C57/BL6 mouse was crossed with an Itgax-cre deleter strain (Caton et al, 2007) to specifically inactivate GADD34 activity in CD11c-expressing cells, including all DC subsets. Despite inducing a light splenomegaly, GADD34 inactivation had no obvious consequences for splenocyte development in vitro and in vivo (Fig S2). Flt3-L BMDCs derived from WT and Itgax-cre/*Ppp1r15a*^loxp/loxp (GADD34ΔC) mice were LPS-activated prior detection of different translation factors by immunoblot (Fig 4A). GADD34 inactivation prevented LPS-dependent eIF2α dephosphorylation; however, phosphorylation levels of the activator β subunit of eIF2 (eIF2β), eEF2, and ribosomal S6 protein remained unchanged, underlining GADD34 specificity for eIF2α (Fig 4A). eIF2β phosphorylation is known to counteract p-eIF2α negative effect and promotes mRNA translation (Gandin et al, 2016). However, in our experimental setting, it was neither impacted by LPS activation nor by the loss of GADD34 activity. eIF2β is, therefore, unlikely to interfere with eIF2α regulation in DCs. Functional deletion of GADD34 inhibited translation initiation in both steady-state and LPS-activated cDC2 (Fig 4B), and also reduced translating polysomes speed in non-stimulated cells. GADD34 expression seems, therefore, to prevent protein synthesis inhibition linked to abundant eIF2α phosphorylation in steady-state DCs. The amount of eIF2B present in the DC seems, however, sufficient to maintain a lower but still active protein synthesis despite GADD34 inactivation and increased p-eIF2α (Fig 4A).

In MEF, whereas induction of GADD34 transcription during ER stress is ATF4 dependent (Walter & Ron, 2011), expression of GADD34 upon viral sensors activation is interferon regulatory factor 3 (IRF3) dependent (Dalet al, 2017). We, therefore, inhibited the TANK-binding kinase 1 (TBK1)/IKKε/IRF3 signaling axis to investigate if it is also responsible of GADD34 induction in LPS-activated DCs. Treatment with the TBK1 inhibitor (MRT67307, TBKin) (Clark et al, 2011) prevented LPS-dependent induction of GADD34 mRNA (Fig 4C). *Ppp1r15a/GADD34* transcription is, therefore, also partially dependent on the TBK1/IKKε signaling cascade in DC and not only on

---

test. **P < 0.01. **(D)** Levels of p-eIF2α and total eIF2α were measured in DC populations by immunoblot. Sorted steady-state Flt3-L BMDCs were compared with MEFs and freshly isolated CD8α+ T cells stimulated or not with thapsigargin (Tg) for 2 h (200 nM). Ratio of p-eIF2α/eIF2α is quantified in the graph of the lower panel. **(E)** Levels of p-eIF2α and total eIF2α were measured in bulk Flt3-L BMDCs during different days of BM differentiation in vitro. **(F)** Levels of protein synthesis were measured by puromycilation and intracellular flow cytometry detection in different subsets of Flt3-L BMDCs and in CD8⁺ splenic T cells. Cells were incubated with puromycin 10 min before harvesting and when indicated, cycloheximide (CHX, 10 μM) was added 5 min before puromycin. Steady-state Flt3-L BMDCs were directly compared with CD8⁺ splenic T cells either steady-state or stimulated overnight with anti-CD3 (10 μg/ml) and anti CD28 (5 μg/ml). Samples without previous incorporation of puromycin were used as control. All data are representative of n = 3 independent experiments. Data in (F) represent mean fluorescence intensity ± SD of three independent experiments. Statistical analysis was performed using unpaired *t* test (****P < 0.0001).

the ATF4-dependent transcriptional axis. Importantly, protein synthesis was reduced in GADD34ΔC cells (Fig 4B) and eIF2α phosphorylation increased upon TBK1 inhibition in resting cells (Fig 4D). However, GADD34 mRNA expression levels were too low to define if basal IKKε/TBK1/IRF3 signaling activity could promote GADD34 mRNA expression in steady-state BMDCs. These results suggest nevertheless that GADD34 transcription and translation is regulated in both steady-state and activated DCs by the IKKε/TBK1/IRF3 signaling cascade (Reid et al, 2016).

## PERK mediates eIF2α phosphorylation in steady-state DCs

We next investigated the consequences of inactivating known eIF2α kinases in steady-state Flt3-L BMDCs (Krishna & Kumar, 2018). We tested pharmacological and genetic inactivation of PKR (EIF2AK2) and GCN2 (EIF2AK4) (Fig S3), without observing any major disturbances in eIF2α phosphorylation levels. We next turned toward the ER-stress kinase PERK (EIF2AK3) by crossing PERK^loxp/loxp mice with the Itgax-cre strain (Caton et al, 2007) allowing for the deletion of the exons 7–9, coding for the kinase domain (PERKΔK) in most CD11c-expressing cells (Fig S1B). PERK protein synthesis levels were enriched in WT CD11c+ splenic DC compared with other splenocytes (Fig 5A). PERK expression was efficiently abrogated in Flt3-L BMDCs and to a relatively lesser extent in spleen DCs isolated from animals bearing the floxed-PERK alleles (Fig 5A). PERK inactivation did not impair DC development in vitro nor in vivo (Fig S4) but decreased p-eIF2α levels by 60% in steady-state DCs (Fig 5B), whereas p-eEF2 levels remained unchanged (Fig 5C). Interestingly, LPS stimulation induced eIF2α dephosphorylation although PERK levels were increased upon activation of WT Flt3-L BMDCs (Figs 4A and 5A). PERKΔK DCs did not display any additional decrease in p-eIF2α levels, suggesting that GADD34/PP1c activity requires functional PERK activity or high p-eIF2α levels to be implemented in DCs. Conversely to GADD34-deficient cells, PERK deletion increased translation initiation and elongation rate as measured by SunRISE in both steady-state and LPS stimulated cDC2 (Fig 5D). PERK is the EIF2AK responsible for most eIF2α phosphorylation in Flt3-L BMDCs DCs and mirrors GADD34 activity to regulate active protein synthesis at steady-state and during DC activation.

## DCs are insensitive to ISR induction by subtilase cytotoxin (SubAB)

PERK is activated during DC development leading to intense eIF2α phosphorylation at steady state, whereas these cells avoid translational arrest, by expressing GADD34 and eIF2B, among other potential compensatory biochemical mechanisms. We initiated a search to identify the cause of PERK activation in DCs by testing if ER stress triggers IRE1α and PERK activation during DC differentiation. We monitored the splicing of XBP1 mRNA that reflects IRE1α pathway activation (Walter & Ron, 2011) and found limited accumulation of the spliced form of the XBP1 mRNA (sXBP1) in the bulk of differentiating DCs (Fig S5A). mRNA expression of other major transcription factors induced during the UPR (Walter & Ron, 2011; Han et al, 2013), such as ATF4 and ATF6, was found moderately increased at day 7, whereas CHOP transcription remained

unaffected during differentiation. Because the induction of these factors is mostly regulated at the posttranscriptional level, we wondered if the constant PERK activity observed in DCs could induce a chronic ATF4-dependent ISR in these cells. Recently, translational and transcriptional programs that allow adaptation to chronic ER stress have been described by Guan et al (2017). This cISR operates via PERK-dependent mechanisms, which allow simultaneous activation of stress-sensing and adaptive responses while allowing recovery of protein synthesis.

We took advantage of available transcriptomic data (GSE9810, GSE2389) and of ATF4/CHOP-dependent gene (Han et al, 2013) to perform a Gene Set Enrichment Analysis (GSEA) and define the level of common gene expression found in DCs and potentially shared with an artificially induced acute or cISR. GSEA was followed by multiple testing correction (Subramanian et al, 2005) using the BubbleGUM software, which allows statistical assessment and visualization of changes in the expression of a predefined set of genes in different conditions (Spinelli et al, 2015). GSEA revealed no significant enrichment of ATF4- and CHOP-dependent genes expression in the DC transcriptome (false discovery rate [FDR] > 0.25) (Fig S6A and B). Acute ISR- and cISR-dependent transcriptions, respectively, obtained after 1- or 16-h treatments with thapsigargin (Guan et al, 2017) were also compared with splenocyte transcriptomes. Again, no significant gene enrichment could be detected during these analyses (Fig S6C and D) (FDR > 0.25).

PERK- and p-eIF2α-mediated translational reprogramming during cISR appears to bypass cap-mediated translation (Guan et al, 2017). We, therefore, tested if protein synthesis in Flt3-L BMDC was independent of 5′ mRNA cap binding eIF4F complex, composed of eIF4A, eIF4E, and eIF4G. We used 4EGI-1, an inhibitor of eIF4F assembly (Moerke et al, 2007), and ROCA, an eIF4A inhibitor (Iwasaki et al, 2019), to treat WT and PERK-deficient DCs and confirm the dependency of their protein synthesis on eIF4F activity. Both compounds had a profound inhibitory effect on DCs translational activity (80% of reduction), irrespective of their subsets or activation state (Fig S5B and C). This level of inhibition indicates that DC mostly depend on eIF4F-dependent cap-mediated translation, thus again contrasting from cells undergoing cISR (Guan et al, 2017). DCs have therefore adapted to the consequences of high eIF2α phosphorylation to allow for translation of their specialized transcriptome, without induction of acute or cISR and ATF4-dependent transcriptional programs.

The lack of ATF4-dependent gene signatures in DCs when compared with other CD45+ cell types made us to wonder whether the high p-eIF2α levels observed at steady state could interfere with DC capacity to respond to ER-stress. ER-chaperone BiP (HSPA5, heat shock protein family A (Hsp70) member 5) is a key component of the UPR. Accumulation of misfolded protein in the ER-lumen causes BiP dissociation from IRE1α and PERK to induce their dimerization and initiate the different signaling cascades controlling the UPR. Flt3-L BMDCs were exposed to subtilase cytotoxin (SubAB), a bacterial AB5 toxin, which by proteolytic cleavage of BiP induces a strong UPR, including PERK-dependent eIF2α phosphorylation (Paton et al, 2006). When WT and PERKΔK DCs were submitted to SubAB treatment, a modest PERK-dependent phosphorylation of eIF2α was observed in WT cells

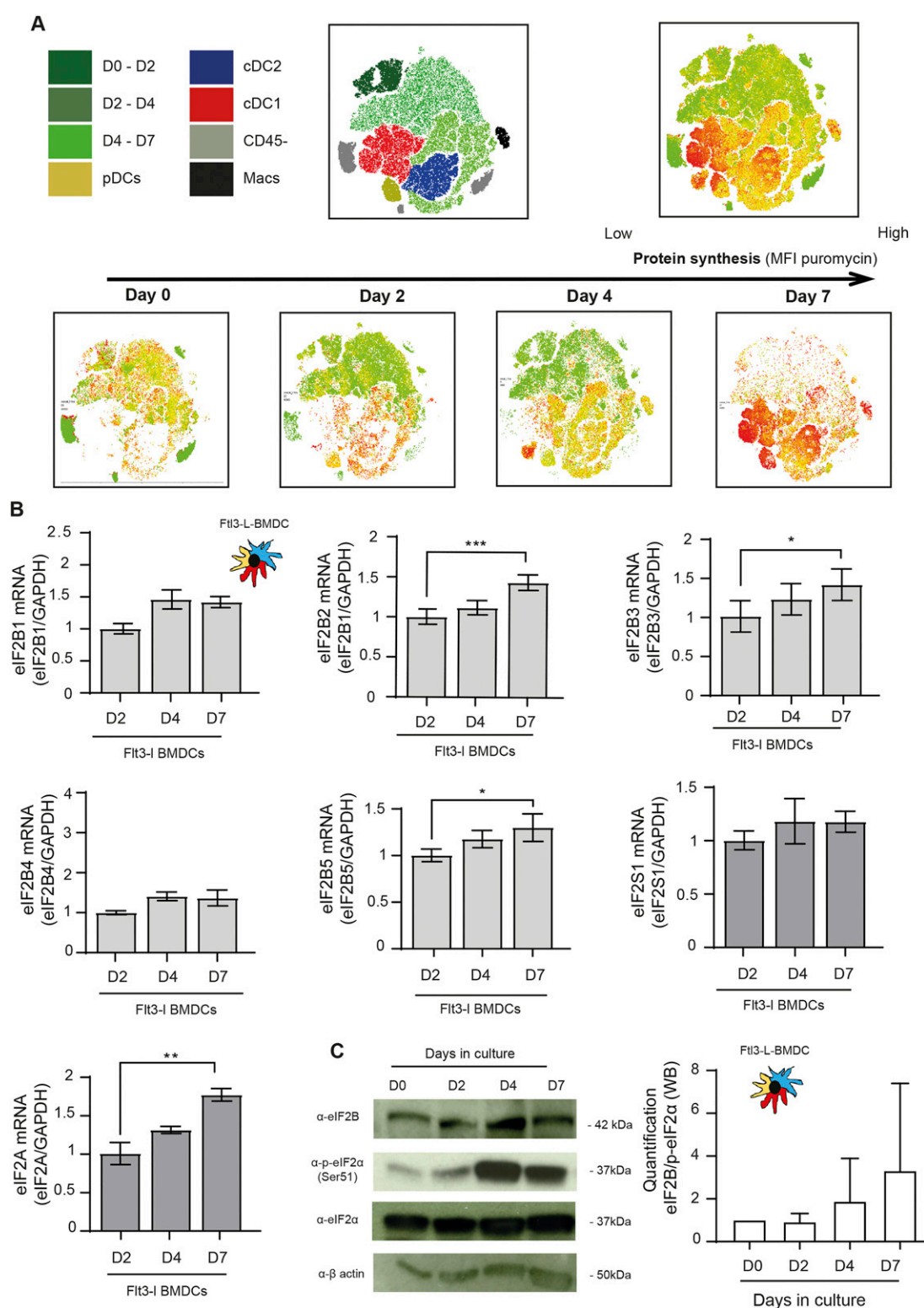

**Figure 2.  Protein synthesis is increased during in BMDCs differentiation.**
**(A)** Levels of protein synthesis were measured every 2 d by flow cytometry during Flt3-L BMDCs differentiation in vitro (0, 2, 4, and 7 d). Cells were incubated during 10 min with puromycin, intracellularly stained with an α-puromycin antibody prior analysis. The same dimensionality reduction using t-distributed stochastic neighbor embeding was applied to all samples. Macrophages (Macs) in black are gated as CD45[+], CD11c+, CD11b+, F4/80+, and CD64[+] cells; cDC1 in red express CD45[+], CD11c+, MCHII+, and CD24[+]; cDC2 in purple express CD45[+], CD11c+, MHC II+, CD11b+, and Sirpα+; pDC in yellow express CD11c[int] and Siglec H+; cells negative for CD45 in gray are considered as non-immune. **(B)** mRNA levels of eIF2Bε (B5); eIF2Bγ (B3), eIF2Bα (B1), eIF2Bβ (B2), eIF2Bδ (B4), eIF2α (eIF2S1), and eIF2A measured by qRT-PCR in bulk Flt3-L BMDCs at

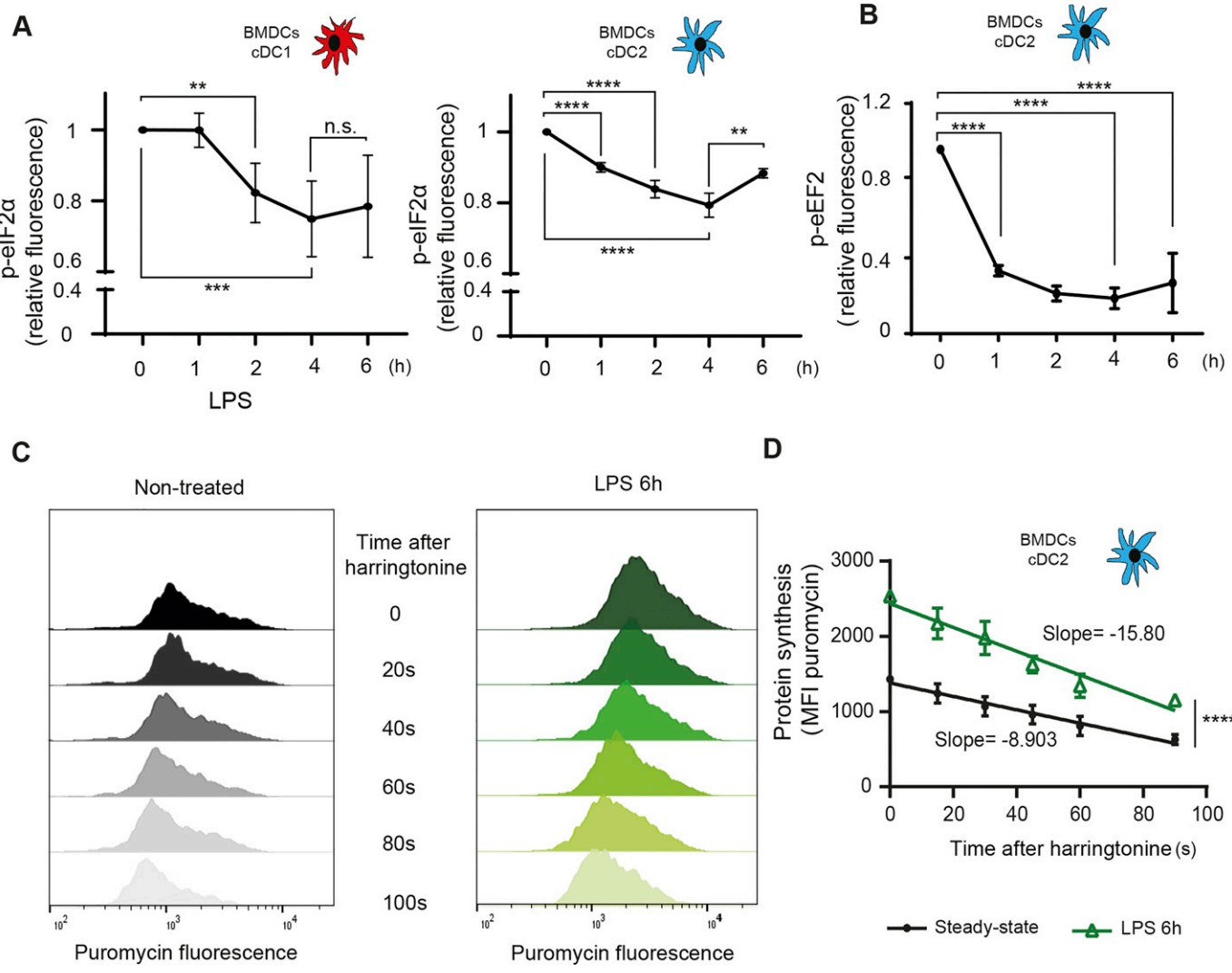

**Figure 3. P-eIF2α and p-eEF2 levels are down-regulated upon LPS stimulation.**
Flt3-L BMDCs were stimulated with LPS (100 ng/ml) for indicated hours. **(A, B)** Monitoring of p-eIF2α and (B) p-eEF2 by intracellular flow in CDC1 and cDC2. **(C)** Flow detection of puromycin incorporation was performed on the cDC2 population. Total puromycin mean fluorescence intensity between steady-state and LPS activated cDC2 is shown for different time of harringtonine treatment. **(D)** Mean fluorescence intensity was plotted as a decay slope and establish the speed of translation elongation. All data are representative of n = 3 independent experiments. Data represent mean ± SD of three independent experiments. **(A, B, C, D)** Statistical analysis was performed using Dunnett's multiple comparison (A, B, C, D) Mann–Whitney test (**$P < 0.01$ and ****$P < 0.0001$).

(Fig 6A), with limited consequences on translation (Fig 6B). In contrast to SubAB, thapsigargin treatment arrested translation more efficiently and triggered stronger eIF2α phosphorylation by PERK, but also by a different EIF2AK because p-eIF2α levels were also increased in PERKΔK cells. Little ATF4 could be detected in the cytosolic or nuclear fractions of control or toxin-treated DC (Fig 6C), reflecting the modest induction of eIF2α phosphorylation (Fig 6D), and confirming the limited impact of SubAB on ISR induction. The efficacy of SubAB treatment was tested in MEFs, in which ATF4 was strongly induced by the toxin and absent from control ATF4−/− cell (Fig 6C). DCs are therefore unable to induce the ISR upon SubAB treatment, despite the activation of other UPR branches, as demonstrated by augmented IRE1α activity (Fig 6E), that is responsible for XBP-1 splicing and translation reduction through IRE1-dependent decay of mRNA (RIDD) (Tavernier et al, 2017). BiP mRNA levels were moderately augmented during DC differentiation. However, the similar expression levels observed for DCs and MEFs (Fig 6F) suggest that BiP transcriptional regulation is not involved in the DC resistance to SubAB. The relatively high PERK and GADD34 activity observed in steady-state DCs, together with

indicated days of differentiation and compared with control MEFs. **(C)** Levels of eIF2Bε, P-eIF2α, total eIF2α, and β-actin were measured in bulk Flt3-L BMDCs at indicated days of differentiation in vitro. Quantification of the ratio eIF2Bε/P-eIF2α is shown on the right. All data are representative of n = 3 independent experiments. Data in (B) represent Mean ± SD of three independent experiments. Statistical analysis was performed using Dunnett's multiple comparison (*$P < 0.05$, **$P < 0.01$, and ***$P < 0.001$).

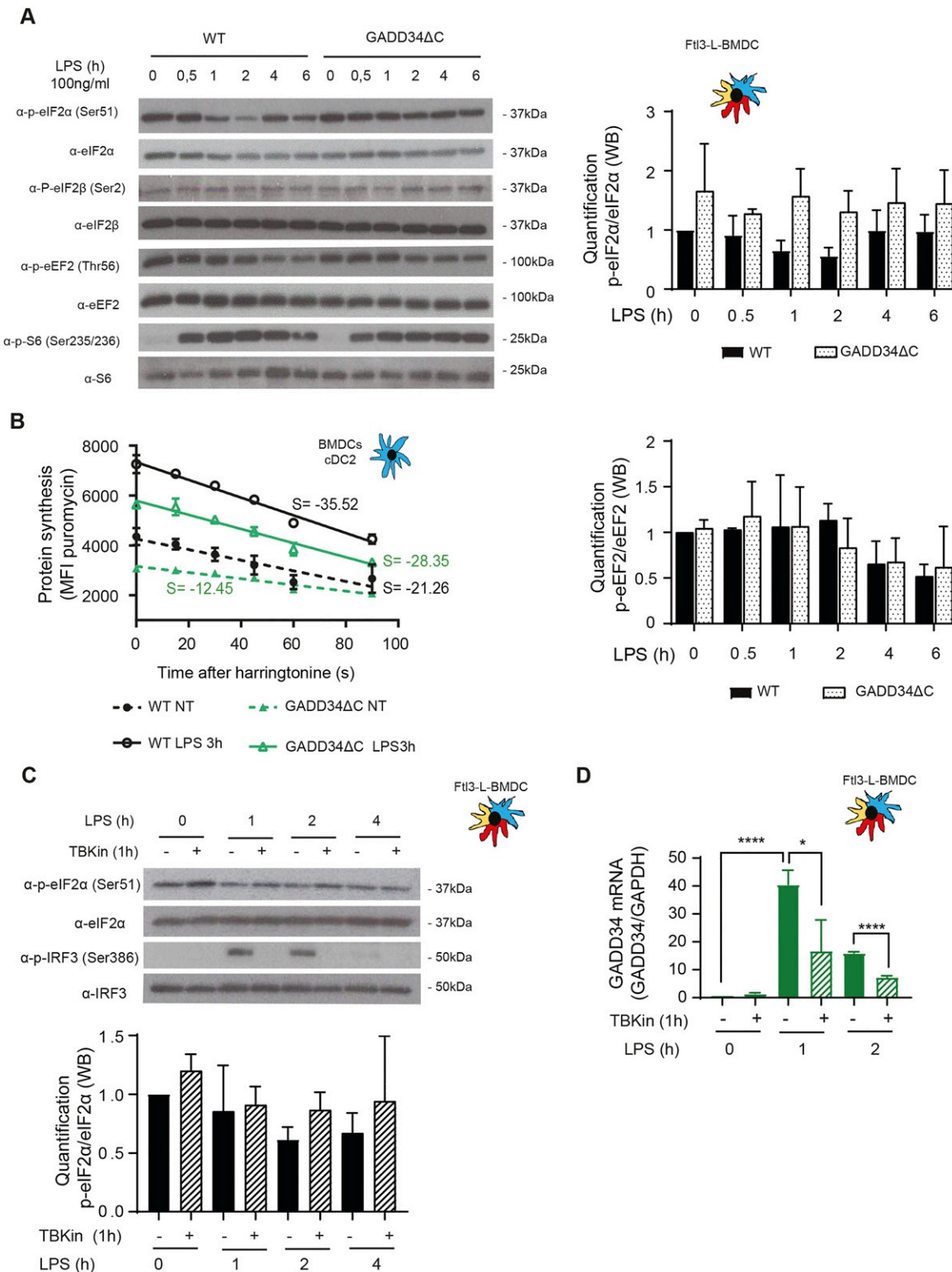

**Figure 4. GADD34 mediates eIF2α dephosphorylation and controls translation in DCs.**
WT and GADD34ΔC BMDCs were stimulated with LPS (100 ng/ml) for the indicated times. **(A)** Levels of p-eIF2α, total eIF2α, p-eIF2β, total eIF2β, p-eEF2, total eEF2, P-S6, and total S6 were detected by immunoblot (top left) and quantification is shown in the different panels. **(B)** The speed of translation elongation was measured by SunRISE after 3 h of incubation with LPS. Harringtonine (2 μg/ml) was added at different times up to 90 s prior incubation with puromycin during 10 min. Flow intracellular staining was performed in cDC2 using an α-puromycin antibody. The total decay of puromycin mean fluorescence intensity between WT and GADD34ΔC in steady-state and upon activation indicates that translation initiation and elongation speed is decreased in GADD34-deficient DCs. Flt3-L BMDCs were pretreated with 2 μM of the TBK1 inhibitor

high eIF2B expression, and potentially eIF2A, would prevent a full ISR induction, including translation arrest, ATF4 synthesis and associated transcriptional response.

## Importance of the ISR for PRRs signaling

In macrophages, p-eIF2α- and ATF4-dependent expression of HSPB8 is required for the assembly of PRR signaling adapters, such as mitochondrial antiviral-signaling protein (MAVS), or TIR domain–containing adapter protein inducing interferon-α (TRIF), but not of myeloid differentiation primary response gene 88 (MyD88). HSPB8 seems necessary for TRIF and MAVS to be incorporated into protein aggregates that constitute signalosomes for different innate immunity signaling pathways triggered by MAMPs (Abdel-Nour et al, 2019). Given the lack of ATF4 synthesis and the resistance of active DCs to mount an acute ISR, we investigated their capacity to produce pro-inflammatory cytokines and type-I IFN in a perturbed eIF2α-phosphorylation context. Importantly, we have previously shown that GADD34-deficient BM-derived and spleen DCs, have a reduced capacity to produce IFN-β (Clavarino et al, 2012b; Perego et al, 2018). IFN-β and IL-6 mRNA expression were therefore quantified after 4 h of LPS activation of WT and PERKΔK DCs. IL-6 transcription (Fig S7A) and secretion (Fig 7A) remained unchanged. However, IFN-β secretion was reduced by half in PERKΔK DCs (Fig 7A), whereas its transcription remained surprisingly unaffected in mutant cells (Fig S7A), suggesting that PERK activity is necessary for normal synthesis and secretion of IFN-β independently of the activation of the TRIF-dependent pathway downstream of TLR4.

We further investigated the importance of eIF2α-phosphorylation with respect to PERK activity using a pharmacological ISR inhibitor (ISRIB) (Sidrauski et al, 2015), which prevents inhibition of eIF2B by p-eIF2α and prevents translation inhibition, as shown here for MEFs in different ISR-inducing conditions (Fig 7B). ISRIB should prevent the induction of the ISR in activated DCs and interfere, as reported for macrophages (Abdel-Nour et al, 2019), with TRIF signaling and down-stream IFN-β expression. In our experimental system, we used LPS and polyinosinic:polycytidylic acid (poly(I:C)) to stimulate, respectively, TRIF-dependent TLR4 and TLR3 (Fitzgerald & Kagan, 2020). IFN-β mRNA induction upon stimulation of DCs with either LPS (Figs 7C and S7B) or poly(I:C) (Fig S7B) was not impaired by ISRIB. IL-6 transcription which is believed to be mostly Myd88-dependent was moderately decreased by ISRIB in LPS-activated DCs and more acutely in BMDM (Fig S7B and C), confirming that cell-specific mechanisms control the transcription of the IL-6 family of cytokines during the ISR (Sanchez et al, 2019). Importantly, TRIF-dependent expression of IFN-β upon LPS activation of BMDM was not impacted by ISRIB treatment, whereas comparatively, poly I:C activation of these cells was too inefficient to obtain statistically reliable data (Fig S7C). We next tested if ISRIB treatment augments IFN-β, IL-6, IL-10, and TNF secretion after 4 h of LPS stimulation in BMDM (Fig 7D). This unchanged (DC) or augmented (BMDM) IFN-β

production observed in the presence of ISRIB confirms that TRIF-dependent signaling does not require acute ISR induction nor ATF4-dependent transcription to promote signalosomes assembly and cytokines expression in DC and probably also macrophages. ISRIB facilitates, however, the production of several cytokines upon activation, confirming that eIF2α phosphorylation decreases the efficacy of cytokines mRNAs translation upon MAMPs detection.

Given the impact of ISRIB on cytokine secretion, we decided to analyze further the response to LPS in cells inactivated for PERK pharmacologically. Cytokines expression was monitored in LPS-activated DCs in presence of the PERK inhibitor GSK2656157 (Axten et al, 2013) (Fig 7C). GSK2656157 treatment decreased both IFN-β and IL-6 secretion by 30–50% (Fig 7A). Over 4 h of treatment, no significant changes in IL-6 mRNA expression was observed, whereas IFN-β transcription was reduced (Fig S7D). These cytokines seem therefore differently affected by alterations in DCs of PERK activity and of eIF2α phosphorylation. IL-6 transcription is sensitive to ISRIB and requires eIF2α phosphorylation. IFN-β transcription and secretion seems, however, dependent on PERK activity, but surprisingly not on eIF2α phosphorylation nor the ISR. We extended our analysis to IL-10 and TNF secretion upon GSK2656157 treatment of LPS-stimulated Flt3L-BMDCs (Fig 8A). These cytokines expression remained, however, unaffected by ISRIB, but like for IFN-β and IL-6, their translation was reduced upon PERK inhibition, suggesting a key role for PERK in promoting cytokines translation in activated DCs.

PERK was recently proposed to control the caspase-1–dependent proteolysis of pro- to mature IL-1β to allow its secretion (Chiritoiu et al, 2019). Given the contrasting effects of PERK inactivation on IFN-β and IL-6 expression, we examined how WT and PERKΔK-DCs co-stimulated with LPS and ATP were promoting the conversion and secretion of mature IL-1β (Fig 8B). Surprisingly, we did not observe any impairment of IL-1β expression in PERKΔK-DCs, but rather an increase by 25% of both IL-1β mRNA transcription and mature IL-1β secretion compared with WT cells (Fig 8B). IL-1β secretion was also monitored in presence of ISRIB and GSK2656157 (Fig 8C), which moderately reduced IL-1β mRNA transcription only in LPS and ATP activating conditions, this without incidence on mature IL-1β secretion. These results suggest that acute pharmacological PERK inactivation has little effect on IL-1β processing and secretion, whereas long-term inactivation favors this secretion potentially by decreasing the inflammasome activation threshold, as previously observed in autophagy deficient macrophages or DCs (Terawaki et al, 2015). PERK inactivation in DCs is therefore not detrimental to IL-1β processing but favors its production and secretion, which could in turn increase IL1B mRNA transcription in a feed-back positive loop (Ceppi et al, 2009).

## Antigen presentation in PERK-deficient DCs

Given the impact of PERK inactivation in DC capacity to secrete type-I IFN, we decided to investigate how it could also interfere with the

---

(MRT67307) for 1 h, before stimulation with LPS (100 ng/ml) for indicated times. **(C)** mRNA levels of GADD34 were measured by qRT-PCR and normalized to the housekeeping gene (GAPDH) level. **(D)** Immunoblot detection of p-eIF2α, total eIF2α, P-IRF3, and total IRF3. Quantification is represented on the right. **(A, B, C)** Statistical analysis was performed using the Wilcoxon test (A, C, B), and Mann–Whitney test (*P < 0.05 and ****P < 0.0001). **(D)** All data are representative of n = 3 independent experiments except (D), n = 2.

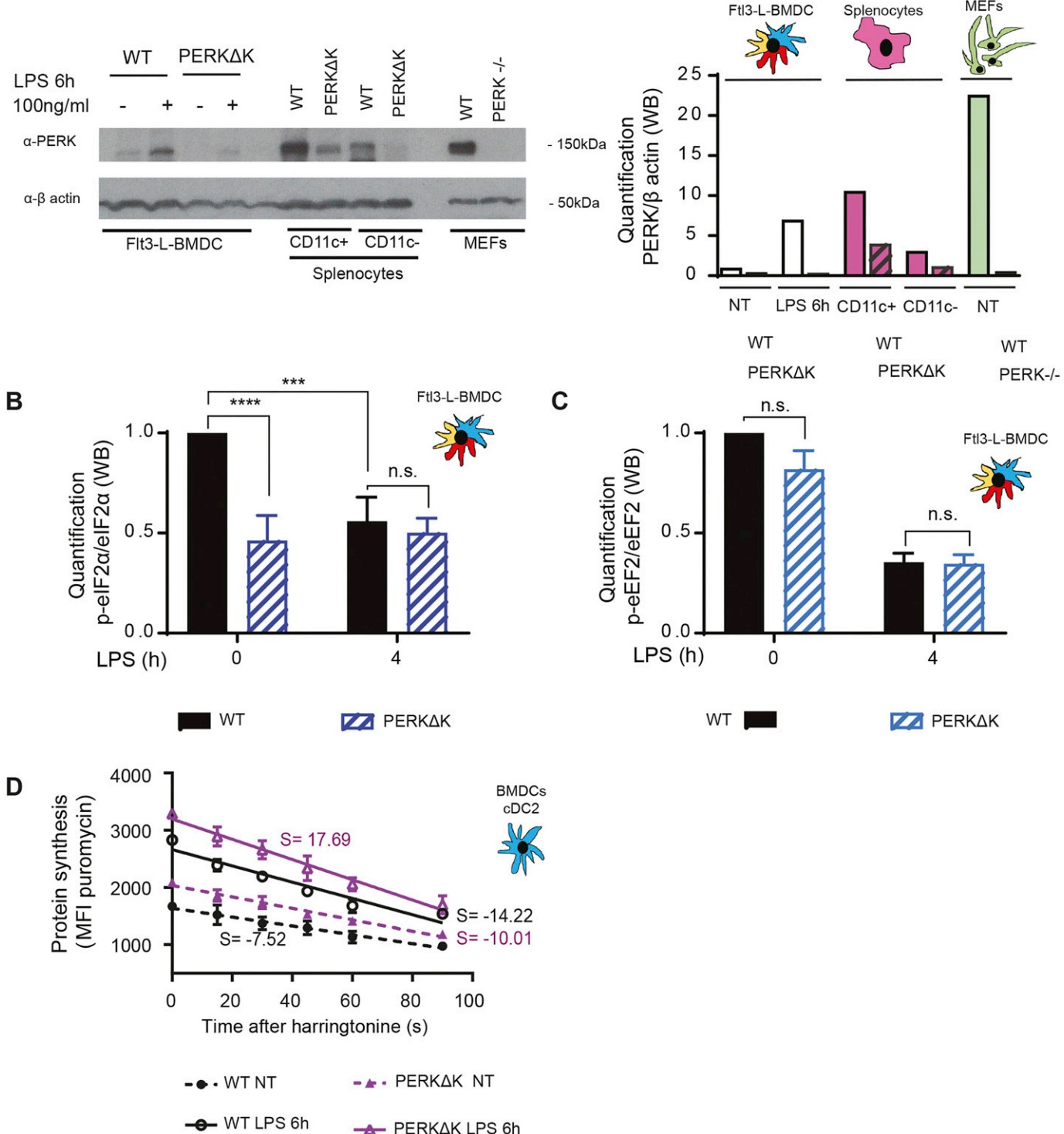

**Figure 5. PERK is activated in steady-state DCs.**
**(A)** Immunoblot detection of PERK and β-actin. Quantification is shown on the right. WT and PERKΔK Flt3-L BMDC treated or not with LPS (100 ng/ml) for 6 h were compared with CD11c+ and CD11c– fractions of splenocytes. WT and PERK–/– MEFs were used as control. **(B, C)** WT and Flt3-L PERKΔK BMDCs were stimulated or not with LPS (100 ng/ml) for 4 h. **(B)** Quantification of p-eIF2α/eIF2α ratio (immunoblot). **(C)** Quantification of p-eEF2/eEF2 ratio (immunoblot). **(D)** The speed of translation elongation was measured using SunRISE in Flt3-L cDC2 after 4 h of incubation with LPS. The total decay of puromycin mean fluorescence intensity between WT and PERKΔK in steady-state and upon activation is showed. All data are representative of n = 3 independent experiments. **(B, C, D)** Statistical analysis was performed using Wilcoxon test (B, C) and Mann–Whitney test (D). Data in (D) represent mean fluorescence intensity ± SD of three independent experiments (**P < 0.01, ***P < 0.001, and ****P < 0.0001).

processing and presentation of exogenous antigens. Surface levels of MHC II and CD86 of WT and PERKΔK BMDC were monitored by flow to establish the capacity of DC1 and DC2 subsets to activate in response to LPS (Fig S8A). Our analysis indicated that WT and PERKΔK BMDC responded equally well to LPS stimulation and did not display differences in surface MHC II nor CD86 levels that could suggest an impairment in their presentation capacity to T cells. We next incubated Flt3-L BMDCs for 8 h with increasing concentrations of hen egg lysozyme (HEL) in presence or not of the PERK inhibitor GSK2656157, prior assaying processing and presentation by measuring CD69 surface up-regulation and the production of IL-2 by the 3A9 (HEL 48-62 on I-Ak) specific T hybridoma (Fig S8B). We found that PERK inhibition had no effect on the efficiency of MHC II restricted processing and presentation of soluble antigens in vitro and consequently did not interfere with the transport of MHC II molecules.

## PERK and actin polymerization coordinates p-eIF2α levels and migration in DCs

Recently, the importance of globular actin in the formation of a tripartite holophosphatase complex assembled with GADD34 and PP1c to dephosphorylate eIF2α was revealed (Chambers et al, 2015; Crespillo-Casado et al, 2017, 2018). Given the unusual regulation of eIF2α phosphorylation in DCs, actin organization could impact this pathway in a different setting than artificial ER stress induction. Actin depolymerizing and polymerizing drugs, respectively, Latrunculin A (Lat A) and Jasplakinoloide (Jaspk) had opposite effects on eIF2α phosphorylation in Flt3-L BMDCs (Fig 9A). globular actin accumulation induced by Lat A was strongly correlated with eIF2α dephosphorylation (Fig 9A and B), whereas actin polymerization induced by Jaspk resulted in a massive increase in eIF2α phosphorylation, together with a reduction in protein synthesis (Fig 9A and C). We next tested the impact of the two drugs on WT and PERKΔK Flt3-L BMDCs activated or not by LPS (Fig 9D). LPS activation or Lat A treatment resulted in the same levels of eIF2α dephosphorylation (Fig 9D). In contrast, Jaspk dominated LPS effect and strongly increased p-eIF2α levels in all conditions tested. PERK inactivation decreased the levels of p-eIF2α, but had no obvious consequences on the efficacy of the drugs because both induced similar responses in WT and PERKΔK cells.

We tested the impact of actin remodeling and translation regulation on the acquisition by DC of their immune-stimulatory phenotypes. Surface MHC II and co-stimulatory molecule CD86 expression were up-regulated by LPS stimulation but remained unaffected by Jaspk treatment (Fig S9A). Similarly, transcription levels of key cytokines such as IL-6 and IFN-β were insensitive to this actin polymerizing drug (Fig S9B and C). However, whereas IL-6 secretion remained identical, IFN-β levels were found reduced by Jaspk treatment, as expected from a situation in which GADD34 activity is reduced (Clavarino et al, 2012b). Interestingly, the IL-6 gene was described to bear an upstream uORF-dependent translational regulation, which could allow IL-6 mRNA translation upon high eIF2α-phosphorylation conditions induced by Jaspk treatment (Sanchez et al, 2019). Taken together, these observations suggest that extensive actin polymerization in DCs increases strongly eIF2α phosphorylation, affecting protein synthesis and

ultimately controlling translationally specific cytokines expression, similarly to what has been observed with Cdc42 or Wiskott–Aldrich syndrome protein mutants (Pulecio et al, 2010; Prete et al, 2013). These results further suggest that actin dynamics and its effect on eIF2α phosphorylation could be key regulating factors for the homeostasis and translation of specific mRNA encoding for proteins generally secreted in a polarized fashion, such as type I IFNs, or associated with cell migration (Pulecio et al, 2010; Prete et al, 2013). Finally, given the interplay between eIF2α phosphorylation and actin polymerization, we wondered whether PERK-deficient cells could display some migratory deficits. We used microfabricated channels, which mimic the confined geometry of the interstitial space in tissues (Heuze et al, 2013; Bretou et al, 2017), to find that PERK-deficient cells were not able to increase their migration speed in response to LPS (Fig 9E), confirming the link between eIF2α phosphorylation and actin dynamics. PERK activity is, therefore, necessary for DCs to acquire normal immune-stimulatory and migratory activities, presumably by coordinating protein synthesis and translation specific mRNAs with actin polymerization.

# Discussion

We have previously proposed the existence of a strong causative link between cell activation by TLR ligands and eIF2α phosphorylation, notably by virtue of strong GADD34 expression in most transcriptomics studies performed on PAMP-activated DCs (Clavarino et al, 2012b; Claudio et al, 2013; Reverendo et al, 2018). Our present work suggests that steady-state DCs activate PERK-mediated eIF2α phosphorylation to acquire their functional properties during differentiation (Fig S8), but distinctly from known ISR programs, normally induced upon acute or chronic ER stress (Han et al, 2013; Guan et al, 2017).

To our knowledge, the level of p-eIF2α observed in primary DC both in vivo and in vitro are unique in their amplitude. Although as judged comparatively from experiments performed with artificial induction of the different EIF2KAs, such p-eIF2α levels should be inhibitory for global protein synthesis (Dalet et al, 2017). DCs have acquired biochemical resistance, like high expression of eIF2B and eIF2A, to compensate for the consequences of this developmental PERK activation and to undergo high eIF2α phosphorylation, whereas maintaining normal proteostasis.

ATF4's role in controlling Ppp1r15a/GADD34 mRNA expression and eIF2α dephosphorylation to restore protein synthesis during stress has been extensively studied (Novoa et al, 2001). Despite high eIF2α phosphorylation, the active translation observed in DCs does not seem to allow ATF4 synthesis and consequently the activation of a bona fide ISR in these cells. In contrast, GADD34 is functional in non-activated DCs with IKKε/TBK1 activity required for its mRNA transcription. This dependency of Ppp1r15a/GADD34 transcription on IKKε/TBK1 confirms that the PPP1R15a gene belongs to a group of genes directly induced by TLR or RLR signaling, as previously suggested by genomic analysis of viral or poly (I:C)-stimulated cells (Freaney et al, 2013; Lazear et al, 2013; Dalet et al, 2017). GADD34 protein expression is undetectable in DCs, without prior treatment with proteasome inhibitors (Clavarino et al, 2012b), which is

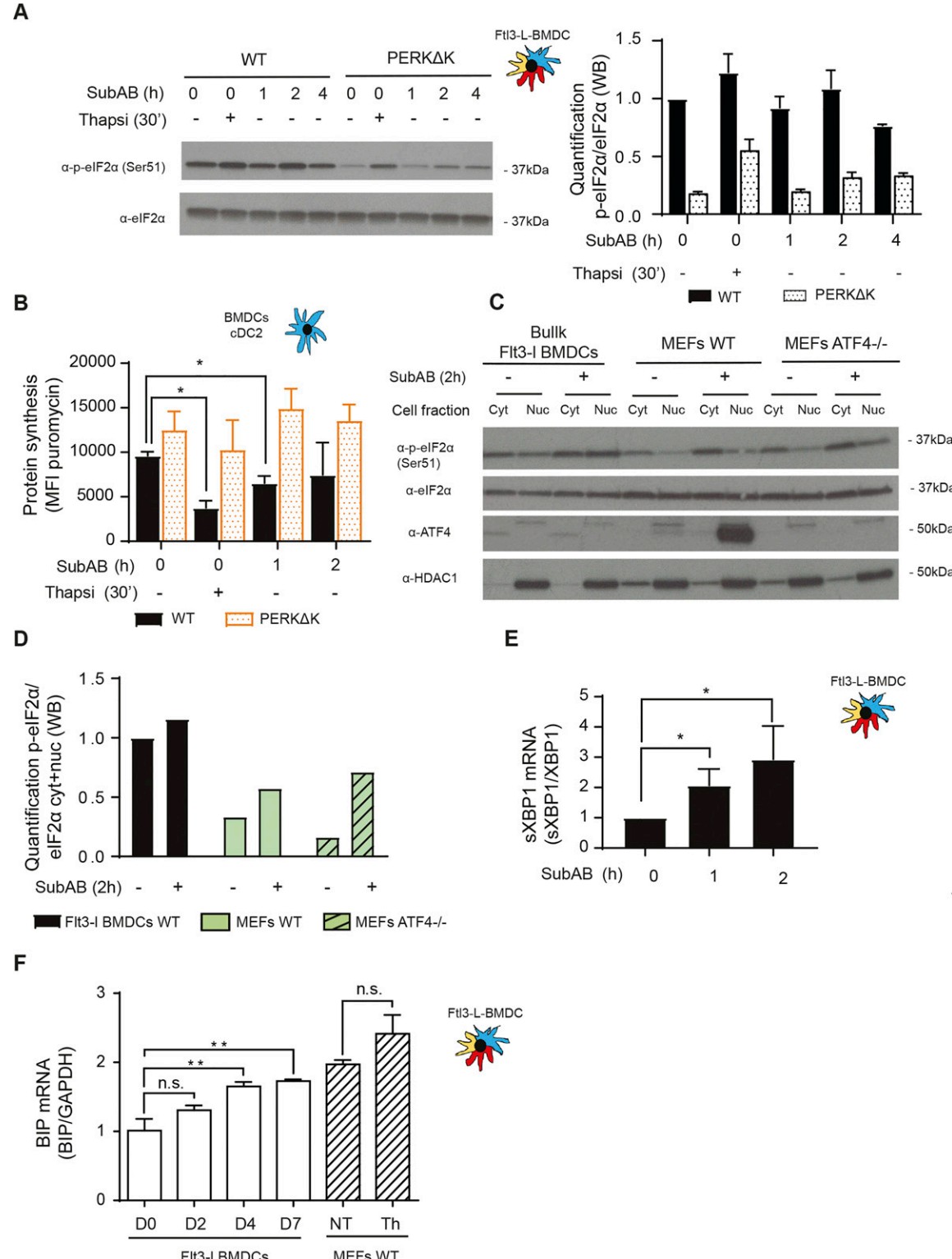

**Figure 6. Flt3-L BMDCs are resistant to subtilase cytotoxin-induced integrated stress response.**

WT and PERKΔK Flt3-L BMDCs were stimulated with thapsigargin (200 nM) and subtilase cytotoxin (SubAB, 250 ng/ml) for the indicated times. **(A)** Levels of p-eIF2α and total eIF2α detected by immunoblot (left) and quantified (right). **(B)** Levels of protein synthesis measured by flow using puromycilation detection in cDC2. Cells were incubated with puromycin 10 min before harvesting. The graph shows the total puromycin mean fluorescence intensity levels. **(C)** WT Flt3-L BMDCs, WT and ATF4−/− MEFs were stimulated with subtilase cytotoxin (SubAB, 250 ng/ml) for 2 h. Both cytoplasmic (cyt) and nuclear (nuc) fractions were analyzed. Levels of p-eIF2α, total eIF2α, ATF4, and HDAC1 (nuclear loading control) were revealed by immunoblot. **(D)** p-eIF2α quantification is represented in (D). **(E)** WT Flt3-L BMDCs were stimulated with subtilase cytotoxin (SubAB, 250 ng/ml) for the indicated times. mRNA levels of spliced XBP1 were measured by qRT-PCR in bulk Flt3-L BMDCs. Raw data were normalized to total XBP1

presumably indicative of an extremely fragile equilibrium between its translation and active degradation. GADD34 mRNA translation, like that of ATF4, is controlled through 5′ upstream ORFs regulation, which are normally bypassed upon general translation arrest to favor the synthesis of these specific ISR molecules (Palam et al, 2011). Interestingly, GADD34 mRNA has been recently shown to be also actively translated in unstressed MEFs, albeit at much lower levels than upon ER stress (Reid et al, 2016). Thus, in steady-state DCs, GADD34 synthesis likely occurs in a high eIF2α-phosphorylation context, despite relatively normal level of translation, whereas that of ATF4 does not. The difference in the 5′ upstream ORFs organization of the mRNAs coding for these two molecules (Palam et al, 2011; Andreev et al, 2015), could explain this difference, and how GADD34/PP1c activity contributes to the maintenance of protein synthesis activity by counteracting PERK in steady-state DCs.

Importantly, the PERK/eIF2α/GADD34 molecular trio sets the physiological range for potential protein synthesis initiation available in the different DC activation stage; however, the upward progression triggered by LPS stimulation, from one level of protein synthesis to the next, does not depend on this biochemical axis and is likely regulated by other protein synthesis regulation pathways, like the mTORC1 or casein kinase 2 pathways (Lelouard et al, 2007; Reverendo et al, 2019) (Fig S10). Other mechanisms that contribute to escape PERK-mediated eIF2α phosphorylation, including the eIF2B-independent and eIF3-dependent pathway recently described to rescue translation during chronic ER stress (Guan et al, 2017), do not seem to be used in the DC context. Given the amount of eIF2A and eIF2B expressed by differentiated DCs, these factors are likely to be sufficient to counteract excessive eIF2α phosphorylation and maintain protein synthesis level in DCs. This activity could be equivalent to the TLR-dependent activation of eIF2B through PP2A-mediated dephosphorylation of the eIF2Bε-subunit, which prevents translation arrest in tunicamycin treated macrophages (Woo et al, 2012).

These DC-specific mechanisms prevent the induction of the ISR by the AB5 subtilase cytotoxin, which targets the ER chaperone BiP. Independently of demonstrating that induction of acute eIF2α phosphorylation by thapsigargin is not solely dependent on PERK activation, our observations suggest that DCs could escape EIF2KA-dependent translation arrest during exposure to different metabolic insults relevant to the immune context. These situations can include viral infection (Clavarino et al, 2012b), exposure to high levels of fatty acids during pathogenesis, oxidative stress during inflammation or amino acids starvation, mediated by amino acid–degrading enzymes, such as arginase 1 or IDO, which are induced during infection or cancer development (Munn et al, 2004; Claudio et al, 2013). Importantly we could also show, which the ISR induction is not necessary for DCs to drive the transcription of pro-inflammatory cytokines in response to TRIF or MAVs dependent-signaling (Abdel-Nour et al, 2019), nor the secretion of IL-1β (Chiritoiu et al, 2019).

PERK activation is therefore required to regulate mRNA translation during DC differentiation and potentially also GADD34 synthesis, which not only provides a negative feed-back to PERK, but also is required for normal DC activation and cytokines expression (Clavarino et al, 2012b; Perego et al, 2018). Although a role for the ISR has been suggested to favor the survival of tissue associated DCs (Tavernier et al, 2017), we could not detect any particular phenotype impairing the development of DCs in the spleen of PERK-deficient animals. A close examination of the functional capacity of DC in vitro showed that although soluble antigen presentation was not affected by PERK inactivation, it induced nevertheless an alteration of IFN-β and cytokines production as well as of DC migratory capacity upon MAMPs activation. Interestingly, ROCK-induced acto-myosin contractility in transformed fibroblasts enhances signaling through PERK and ATF4 (Boyle et al, 2020), whereas PERK itself has been shown to interact with filamin-A and to participate to F-Actin remodeling in MEFs (van Vliet et al, 2017), suggesting that the migratory deficit observed in PERK-deficient DCs could be also dependent on these interactions. This finding echoes with the existence of a cross-talk between actin skeleton organization and the main molecular actors involved in the ISR (Chambers et al, 2015; Chen et al, 2015). We confirmed that globular actin synergizes with the PP1c to dephosphorylate p-eIF2α, suggesting that the PERK/GADD34 pathway could play an important role in regulating translation in response to actin dynamics and possibly in coordinating migration or interactions with T cells.

DCs therefore represent a model of choice for studying this possibility, given their developmental regulation of eIF2α phosphorylation and their requirement for actin dependent-phagocytosis and migration to perform their immune-stimulatory function. The activation of PERK/GADD34 pathway in steady-state DCs also underlines the importance of these molecules in homeostatic condition, independently of obvious acute ER stress, for the acquisition of specialized function. Clearly, the use of PERK-deficient cells versus pharmacological inhibition creates some discrepancies on how several biochemical functions in DCs are truly affected directly by PERK loss or reduced eIF2α phosphorylation. Our findings open nevertheless new pharmacological perspectives for therapeutic immune intervention by targeting PERK, GADD34, or eIF2α phosphorylation.

# Materials and Methods

### Cell culture

BM was collected from 6- to 9-wk-old female mice and differentiated in DCs or macrophages during 7 d. The culture was kept at 37°C, with 5% $CO_2$ in Roswell Park Memorial Institute medium (RPMI) (GIBCO), 10% FCS (Sigma-Aldrich), 100 U/ml penicillin, 100 U/ml streptomycin (GIBCO), and 50 μM β-mercaptoethanol (VWR) supplemented with Flt3-L, produced using B16-Flt3-L hybridoma cells for DC differentiation or M-CSF for macrophages, as described previously (Wang et al, 2013). For the migration assays, GM-CSF was used instead of Flt3-L and cells were cultured during 10–12 d with

mRNA expression. **(F)** Levels of BiP mRNA expression measured by qRT–PCR during Flt3-L BMDCs differentiation and in MEFs stimulated with thapsigargin for 30 min. Data are mean ± SD (n = 3). Statistical analysis was performed using Dunnett's multiple comparison (*$P < 0.05$ and **$P < 0.01$).

changes in the medium each 3 d. GM-CSF was obtained from transfected J558 cells (Pierre et al, 1997). To obtain splenocytes, spleens were collected and injected with Liberase TL (Roche) and incubated 25 min at 37°C to disrupt the tissues. DC purification was performed using a CD11c+ positive selection kit (Miltenyi), according to the manufacturer's instructions and CD8α+ T-cell isolation was performed with a Dynabeads untouched mouse CD8 T cells kit from Thermo Fisher Scientific. CD8α+ T where incubate overnight with anti-CD3 (10 μg/ml) and anti-CD28 (5 μg/ml) antibodies, to mimic activation by APCs. MEFs used in this work, ATF4−/− and matched WT (129 SvEv) were a kind gift from Prof. David Ron (Cambridge Institute for Medical Research). PERK KO−/− and matched WT were a kind gift from Prof. Douglas Cavener (Penn State University). MEFs were cultured in DMEM medium (GIBCO) with 5% FBS (Sigma-Aldrich) and 50 μM 2-mercaptoethanol. For the experimental assays, cells were plated from 16 to 24 h before stimulation in six well plates, at 150,000 cells/ml in 2 ml of the same medium. After stimuli, cells were treated with trypsin−EDTA for 2 min at 37°C before washing to detach cells from the wells.

### Reagents

LPS (*E. coli* O55:B5), cycloheximide, puromycin, MRT67307, GSK2656157, rocaglamide, and thapsigargin were purchased from Sigma-Aldrich. Harringtonine is from ABCAM, Latrunculin A, and Jasplakinolide are from Merck-Millipore. Low molecular weight polyinosinic-polycytidylic acid (LMW poly(I:C)) was from InvivoGen. SAR1 was kindly provided by Sanofi and Integrated Stress Response Inhibitor (ISRIB) was a gift from Carmela Sidrausky and Peter Walter (UCSF). Subtilase cytotoxin (Shiga toxigenic *E. coli* strains) was purified from recombinant *E. coli*, as previously described (Paton et al, 2004). 4EGI-1 was purchased by Bertin bioreagent. HEL and the peptide HEL 46-61 were purchased from Thermo Fisher Scientific.

### Flow cytometry analysis

Cell suspensions were washed and incubated with a cocktail of coupled specific antibodies for cell surface markers in flow activated cell sorting (FACS) buffer (PBS, 1% FCS, and 2 mM EDTA) for 30 min at 4°C. For Flt3-l BMDCs, the antibodies used were CD11c (N418), SiglecH (551), CD86 (GL-1), F4/80 (BM8), CD64 (X54-5/7.1) from BioLegend; Sirpα (P84), CD24 (M1/69), and MHC II (M5/114.15.2) from eBioscience CD11b (M1/70) from BD Bioscience. For splenic cells, the antibodies used were NKp46 (29A1.4), CD4 (RM4-5), CD3 (145-2C11), CD11c (N418), CD19 (eBio1D3), CD8α (53-6.7) from eBioscience, BST2 (927), Ly6G, F4/80, Ly6C from BioLegend; CD11b (M1/70), B220 (RA3-6B2), and CD69 (H1.2F3) from BD Biosciences. These antibodies were used in combination with the LIVE/DEAD Fixable Aqua Dead Cell Stain (Thermo Fisher Scientific). For intracellular staining, cells were next fixed with BD Phosflow Fix Buffer I (BD Biosciences) during 10 min at room temperature and washed with 10% Perm/wash Buffer I 1× (BD Biosciences). Permeabilized cells were blocked during 10′ with 10% Perm/wash buffer 1×, 10% FCS, before staining with primary antibodies. When the primary antibody was not coupled, cells were washed after and blocked during 10 min with Perm/wash buffer 1×, 10% FCS, and 10% of serum from the species where the secondary antibody was produced. Then, the incubation with the secondary antibody was performed at 4°C during 30 min. p-eIF2α(S51) was purchased from ABCAM and

p-eEF2(Thr56) from Cell Signaling and Deoxyribonuclease (DNAse I) was purchased from Invitrogen. Data were acquired on an LSR-II/UV instrument using FACS Diva software. The acquired data were analyzed with FlowJo software (BD Biosciences).

### Translation intensity and speed measurement

SUnSET technique to measure the intensity of protein synthesis was used as previously described (Schmidt et al, 2009). Puromycin was added in the culture medium at 12.5 μg/ml, and the cells were incubated for 10 min at 37°C and 5% $CO_2$ before harvesting. Cells were washed with PBS before cell lysis and immunoblotting with the anti-puromycin 12D10 antibody (Merck Millipore). For flow cytometry (flow) cells were processed, as described below for the intracellular staining, using the α-puromycin 12D10 antibody directly conjugated with Alexa 488 or A647 from Merck Millipore. The SUnRISE technique was performed as described (Arguello et al, 2018). Samples were treated with 2 μg/ml of harringtonine at different time points (90, 60, 45, 30, 15, and 0 s) and then treated for 10 min with 12.5 μg/ml of puromycin. For the measurement of Cap-dependent translation, the cells were treated with 4EGI-I (100 μM) or rocaglamide (RocA-1) (100 nM) for 0.5, 1, 2, or 4 h. Cells were then incubated for 15 min at 37°C and stained with the 12D10 antibody (Merk-Millipore).

### Gene expression analysis

Total RNA was extracted from the DCs using the RNeasy Mini Kit (QIAGEN), including a DNA digestion step with RNAse-free DNAse (QIAGEN), and cDNA was synthesized using the Superscript II Reverse Transcriptase (Invitrogen). Quantitative PCR amplification was performed using SYBR Green PCR master mix (Takara) using 10 ng of cDNA and 200 nM of each specific primer on a 7500 Fast Real-PCR system (Applied Biosystems). cDNA concentration in each sample was normalized to GAPDH expression. The primers used for gene amplification were the following: GADD34 (S 5′-GACCCCTCC AACTCTCCTTC-3′, AS 5′-CTTCCTCAGCCTCAGCATTC-3′); IL-6 (S 5′-CAT GTTCTCTGGGAAATCGTG-3′, AS 5′-TCCAGTTTGGTAGCATCCATC-3′); IFN-β (S 5′-CCCTATGGAGATGACGGAGA-3′, AS 5′-ACCCAGTGCTGGAGAAATTG-3′); IL-12 (S 5′-GGAATGTCTGCGTGCAAGCT-3′, AS 5′-ACATGCCCACTTGCTGCAT-3′); ATF4 (S 5′-AAGGAGGATGCCTTTTCCGGG-3′, AS 5′-ATTGGGTTCACT GTCTGAGGG-3′); CHOP (S 5′-CACTTCCGGAGAGACAGACAG-3′, AS 5′-ATGA AGGAGAAGGAGCAGGAG-3′); PERK (S 5′-CGGATTCATTGAAAGCACCT-3′, AS 5′-ACGCGATGGGAGTACAAAAC-3′); XBP1 (S 5′-CCGCAGCACTCAGACTATG-3′, AS 5′-GGGTCCAACTTGTCCAGAAT-3′); spliced XBP1 (S 5′-CTGAGT CCGCAGCAGGT-3′, AS 5′-AAACATGACAGGGTCCAACTT-3′); GAPDH (S 5′-TGGAGAAACCTGCCAAGTATG-3′, AS 5′-GTTGAAGTCGCAGGAGACAAC-3′); IL1-β (S 5′-TGATGTGCTGCTGCGAGAGATT-3′, AS 5′-TGCCATTTTGACAGTGA-3′); eIF2Bε (S 5′-GAGCCCTGGAGGAACACAGG-3′ AS 5′-CACCACGTTGT CCTCATGGC-3′); BIP S (5′-ATTGGAGGTGGGCAAACCAA-3′ AS 5′-TCGCTG GGCATCATTGAAGT-3′).

### GSEA

The GSEA was performed using published murine microarray datasets accessible through the Gene Expression Omnibus repository under the references GSE9810 (Robbins et al, 2008) and

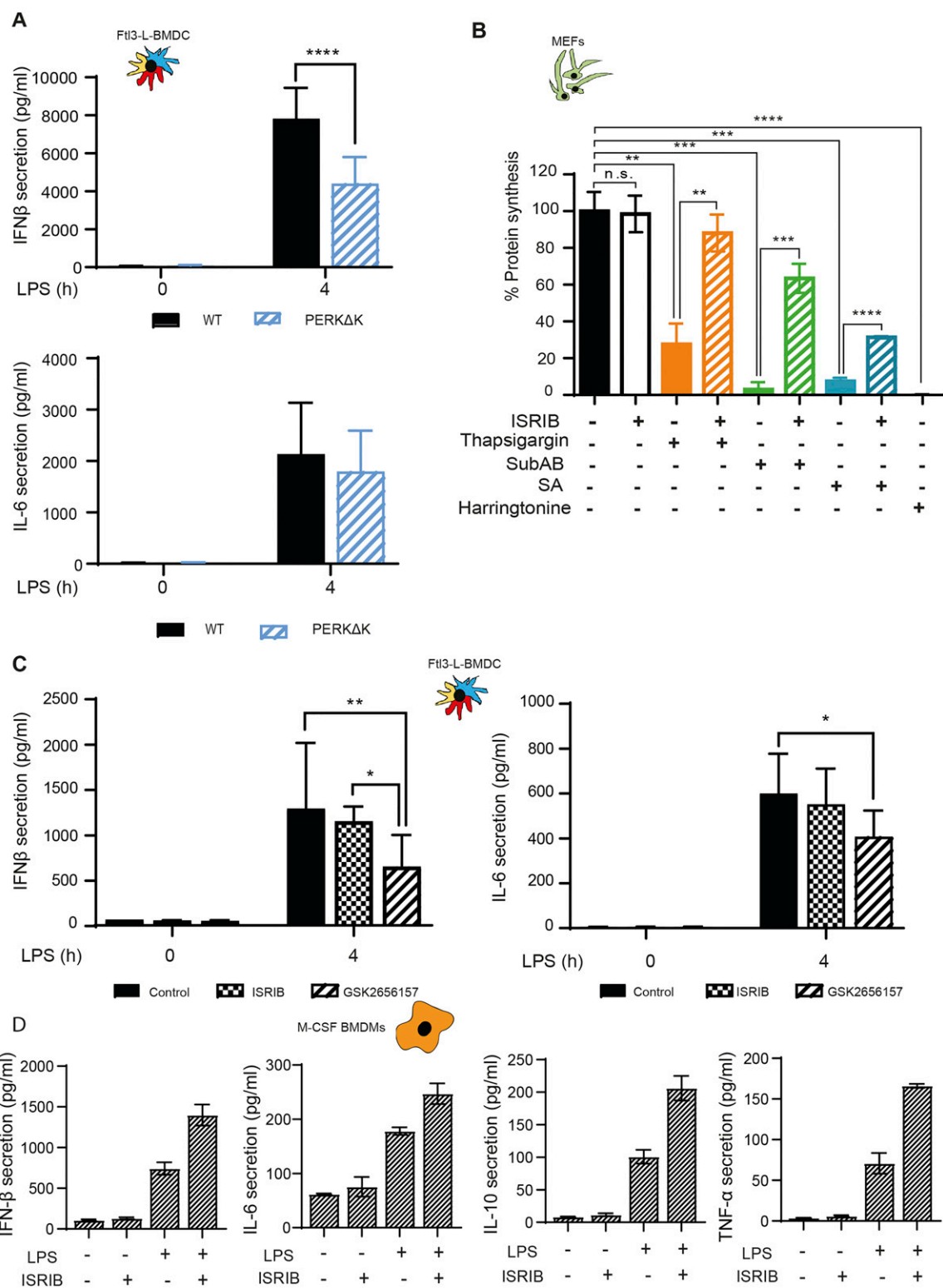

**Figure 7. ISRIB does not inhibit cytokines expression in LPS activated DCs and Macrophages.**
**(A)** IFN-$\beta$ and IL-6 secretion was measured by Legendplex in WT and PERK$\Delta$K Flt3-L BMDCs stimulated with LPS during 4 h. **(B)** Protein synthesis was measured by puromycilation and flow in MEFs treated with ISRIB and different the integrated stress response inducing drugs, thapsigargin (Tg), subtilase cytotoxin (SubAB), and sodium arsenite for indicated times. **(C, D)** IFN-$\beta$, IL-6, IL-10, and TNF secretion was measured by Legendplex in (C) Flt3-L BMDCs and (D) M-CSF BMDM. Data are mean ± SD (n = 3). Statistical analysis was performed using Wilcoxon test (*P < 0.05 and **P < 0.01).

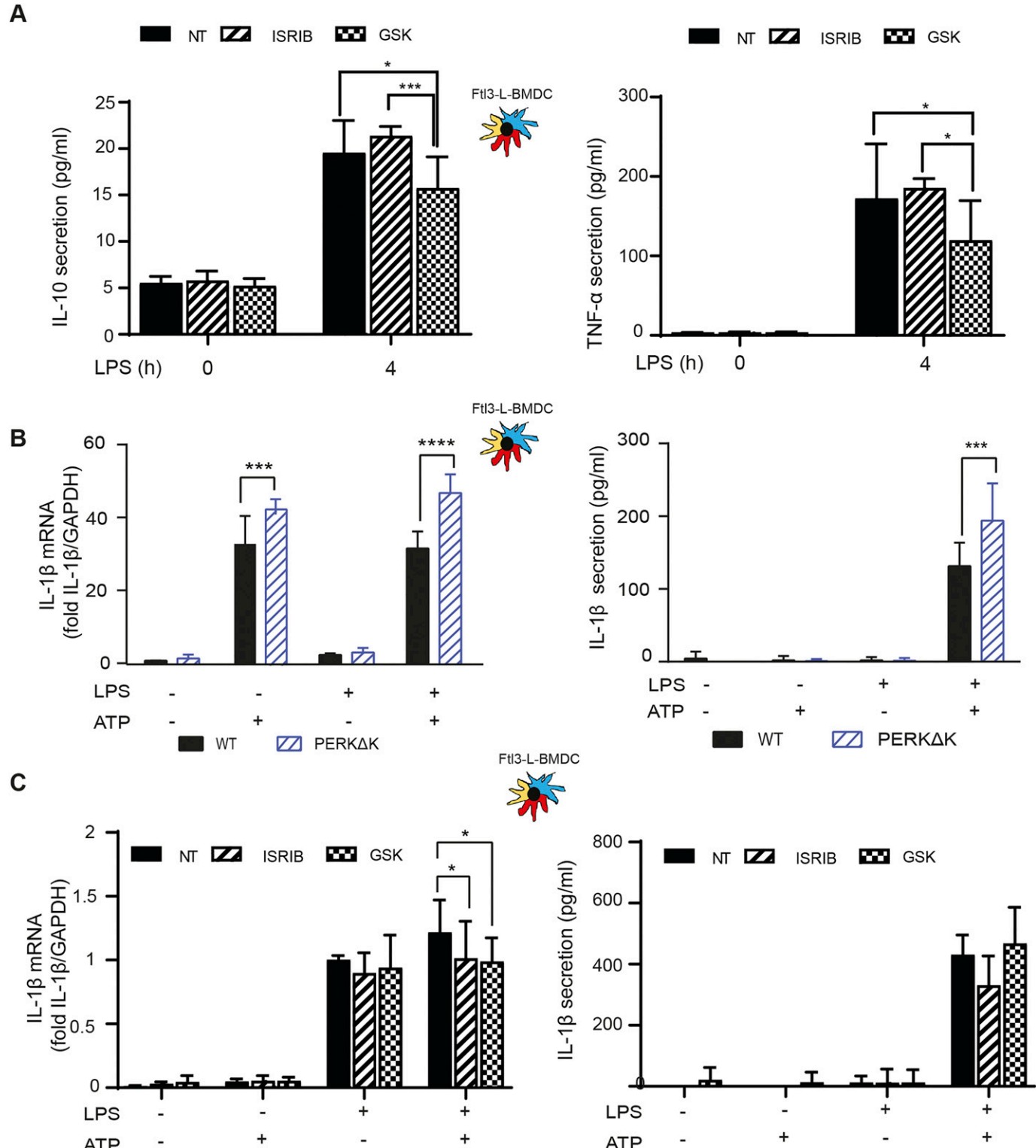

**Figure 8. PERK inactivation and cytokines expression.**
**(A)** IL-10 and TNF secretion was measured by Legendplex in Flt3-L BMDCs activated with LPS and treated or not with ISRIB or GSK2656157 for 4 h. **(B)** IL-1$\beta$ mRNA expression measured by qRT-PCR (left) and secretion by ELISA (right) in WT and PERK$\Delta$K Flt3-L BMDCs stimulated with LPS during 4 h and with ATP for the last 30 min of treatment. **(C)** IL-1$\beta$ mRNA expression measured by qRT-PCR (left) and secretion by ELISA (right) in Flt3-L BMDCs activated with LPS and treated with GSK2656157 and/or ISRIB for 4 h and with ATP for the last 30 min of treatment. **(A, B, C)** Statistical analysis was performed using Dunnett's multiple comparison (A, B) and Wilcoxon test (C). Data are mean ± SD (n = 3). independent experiments (*$P < 0.05$, **$P < 0.01$, and ***$P < 0.001$).

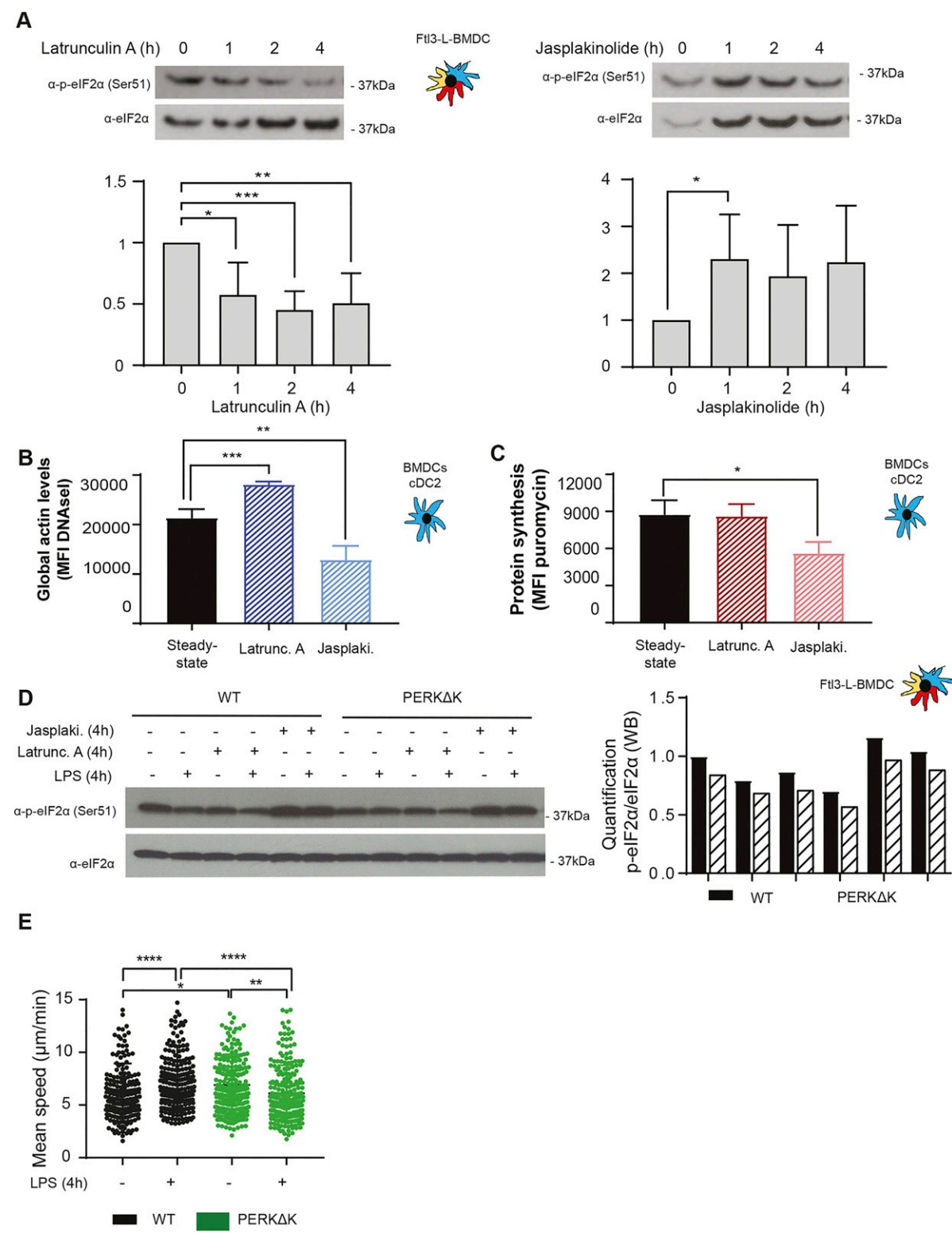

**Figure 9. eIF2α phosphorylation levels are regulated by G-actin availability.**
WT Flt3-L BMDCs were treated with Latrunculin A (50 nM) (Latrunc A) and Jasplakinolide (1 μM) (Jasp) for the indicated times. **(A)** Levels of p-eIF2α and total eIF2α detected by immunoblot in Flt3-L BMDCs. **(B)** Globular actin levels were measured by flow cytometry intracellular staining using a fluorochrome coupled DNAse I protein in cDC2 population from Flt3-L BMDCs. The graph shows total mean fluorescence intensity levels. **(C)** Levels of protein synthesis in cDC2 were measured by puromycilation and flow intracellular staining. Cells were incubated with puromycin 10 min before harvesting. The graph shows total puromycin mean fluorescence intensity levels. WT and PERKΔK Flt3-L BMDCs were treated with LPS (100 ng/ml), Latrunc A, and Jasp for 4 h. **(D)** Levels of p-eIF2α and total eIF2α monitored by immunoblot in Flt3-L BMDCs.

GSE2389 (Fontenot et al, 2005). Raw microarray data describing CD8∝ + cDCs (DC1), CD11b+ cDC (DC2), pDC (plasmacytoid DCs), B cells, NK cells, CD8[+] T cells (Robbins et al, 2008), and CD4[+] T cells (Fontenot et al, 2005) in mice spleen were downloaded. For each of these cell types, the hybridization was performed using Affymetrix mouse 4302.0 gene chips. Microarray data were normalized by Robust Multi-array Average algorithm (Irizarry et al, 2003) using the oligo BioconductorR package (Carvalho & Irizarry, 2010). Normalization consists of a background correction of raw intensities, a $log_2$ transformation followed by quantile normalization to allow the comparison of each probe for each array. Before data usage, the absence of batch effect was assessed by Principal Component Analysis using ade4R package (Bougeard & Dray, 2018).

GSEA was performed using publicly available gene signatures reflecting an ISR state (Tables S1–S3). Lists of ATF4 and CHOP target genes identified by ChIP-seq experiments (Han et al, 2013) have been used to search for ATF4-dependent and CHOP-dependent signatures in DCs (Tables S2 and S3). Lists of genes for which a translational up-regulation after 1 h of thapsigargin (Tg) treatment and congruent (both transcriptional and translational) up-regulations after 16 h of Tg treatment (Guan et al, 2017) were used to search gene expression signatures, respectively, of acute and/or cISRs in DCs. GSEA was generated using BubbleGUM (Spinelli et al, 2015). Briefly, GSEA pairwise comparisons are performed for each probe and the multiple testing effects are corrected using a Benjamini–Yekutieli procedure. The corrected P-values are hence calculated based on a null hypothesis distribution built from the permutations of the gene sets across all the pairwise comparisons. In our analyses, 10,000 permutations of the gene sets have been performed to compute the P-values. All results with a FDR below the threshold of 0.25 have been considered as significant.

## Immunoblotting

Cells were lysed in RIPA buffer (25 mM Tris–HCl, pH 7.6, 150 mM NaCl, 1% NP-40; 1% sodium deoxycholate, 0.1% SDS) supplemented with Complete Mini Protease Inhibitor Mixture Tablets (Roche), NaF (Ser/Thr and acidic phosphatase inhibitor), $Na_3VO_4$ (Tyr and alkaline phosphatase inhibitor) and MG132 (proteasome inhibitor). The nuclear extraction was performed using the Nuclear Extract kit (Active Motif) according with manufacturer's instructions. Protein quantification was performed using the BCA Protein Assay (Pierce). Around 20 µg of soluble proteins were run in 4–20% acrylamide gradient gels and for the immunoblot the concentration and time of incubation had to be optimized for each individual antibody. Rabbit antibodies against eIF2α, p-eEF2(Thr56), eEF2, eIF2B, p-IRF3 (ser396), IRF3, p-S6, and PERK were purchased from Cell Signaling (ref 5324, 2331, 2332, 3592, 4947, 4302, 2211, and 3192, respectively). Rabbit antibody against p-eIF2α(S51) was purchased from ABCAM (Ref 32157). Rabbit antibody against ATF4 was purchased from Santa Cruz

Biotechnology (sc-200). Mouse antibody against β-actin was purchased from Sigma-Aldrich (A2228). Mouse antibodies against HDAC1 and S6 were purchased from Cell Signaling (ref 5356, 2317, respectively). Mouse antibody against puromycin was purchased from Merck Millipore (MABE343). Mouse antibody against p-eIF2β was a kind gift from David Litchfield (University of Western Ontario). Mouse antibody against eIF2β was purchased from Santa Cruz Biotechnology (sc-9978). HRP secondary antibodies were from Jackson ImmunoResearch Laboratories.

## Cytokine measurement

The IL-6, IFN-β, IL1-β, and IL-2 quantifications from the cell culture supernatant were performed using the mouse Interleukin-6 ELISA Kit (eBioscience), the mouse IFN-β ELISA Kits (PBL-Interferon Source or Thermo Fisher Scientific), the mouse IL1-β, and mouse IL2 uncoated ELISA kits (Invitrogen) according to the manufacturer's instructions. Cytokine monitoring was also performed using Legendplex 740150 (BioLegend).

## Antigen presentation assays

Flt3-L-differentiated bmDC obtained from C3H/HeN were treated with indicated concentration of HEL or with 5 µM of 46-61 HEL peptide and incubated for 8 h with 100 nM of LPS in presence or not of GSK2656157. DCs were fixed mildly with 0.25% PFA, 2 min, RT, and prior quenching with 10 mM glycine. DCs were co-cultivated with 3A9 (HEL 46-61 on I-Ak) specific T hybridoma at 5:1 (T:DC) ratio for 18 h. CD69 up-regulation and IL-2 production was determined, respectively, by cytometry and ELISA.

## Immunohistochemistry

Spleens were snap frozen in Tissue Tek (Sakura Finetek). Frozen sections (8 µm) were fixed with acetone permeabilized with 0.05% saponin. The following antibodies were used for the staining: CD11c (N418) from BioLegend (Ref 117301), p-eIF2α (Ser 52) from Invitrogen (Ref 44-728G), CD11b (M1/70) from BD Biosciences, CD8α-biotin (53-6.7) BioLegend (Ref 100703), and B220 (RA3-6B2) from Invitrogen (Ref 14-0452-81). Images were collected using a Zeiss LSM 510 confocal microscope. Image processing was performed with Zeiss LSM software.

## Mice

Wild-type (WT) female C57BL/6 and C3H/HeN mice were purchased from Janvier. PKR−/− C57BL/6 were a kind gift from Dr Bryan Williams (Hudson Institute of Medical Research) (Kumar et al, 1997). PERK[loxp/loxp] mice were the kind gift of Dr Doug Cavener (Zhang et al, 2006) and purchased from Jackson Laboratories. GADD34ΔC[loxp/loxp] mice were developed at the Centre d'Immunophénomique (CIPHE).

---

Quantification is shown on the right. **(E)** WT and PERKΔK GM-CSF BMDCs were treated with LPS (100 ng/ml) d—uring 30 min previous to 16 h of migration. The graph represents instantaneous mean velocities of migration in 4 × 5-µm fibronectin-coated microchannels of at least 100 cells per condition. All data are representative of n = 3 independent experiments. **(A, B, C)** Statistical analysis was performed using Dunnett's multiple comparison (A) and Mann–Whitney test (B, C). Data are mean ± SD (n = 3). *P < 0.05, **P < 0.01, ***P < 0.001, and ****P < 0.0001.

PERK$^{loxp/loxp}$ and GADD34$^{loxp/loxp}$ were crossed with Itgax-Cre+ mice (Caton et al, 2007) and backcrossed, to obtain stable homozygotic lines for the loxp sites expressing Cre. For all studies, age-matched WT and transgenic 6–9 wk females were used. All animals were maintained in the animal facility of CIML or CIPHE under specific pathogen–free conditions accredited by the French Ministry of Agriculture to perform experiments on live mice. These studies were carried out in strict accordance with Guide for the Care and Use of Laboratory Animals of the European Union. All experiments were approved by the Comité d'Ethique PACA and MESRI (approval number APAFIS#10010-201902071610358). All efforts were made to minimize animal suffering.

### Preparation of microchannels and speed of migration measurement

Microchannels were prepared as previously described (Vargas et al, 2016). For velocity measurements (carried out in 4-by-5 μm microchannels), phase-contrast images of migrating cells were acquired during 16 h (frame rate of 2 min) on an epifluorescence Nikon Ti-E video microscope equipped with a cooled charge-coupled device camera (HQ2; Photometrics) and a 10× objective. Kymographs of migrating cells were generated and analyzed using a custom program.

### Statistical analysis

Statistical analysis was performed using GraphPad Prism Software. The most appropriate statistical test was chosen according to each data set. Mainly, we used Wilcoxon test, Mann–Whitney test, *t* test, and multiple comparison with Dunnett's correction. *$P$ < 0.5, **$P$ < 0.01, ***$P$ < 0.001, and ****$P$ < 0.0001.

## Supplementary Information


## Acknowledgements

We thank all the Centre d'Immunologie de Marseille-Luminy (CIML) cytometry and Imaging core facilities for expert assistance. The laboratory is supported by grants from La Fondation de l'Association pour la Recherche sur le Cancer (ARC). The laboratory is "Equipe de la Fondation de la Recherche Médicale" (FRM) sponsored by the grant DEQ20140329536. The project was also supported by grants from l'Agence Nationale de la Recherche (ANR), « ANR-FCT 12-ISV3-0002-01» and « INFORM Labex ANR-11-LABEX-0054 », «DCBIOL Labex ANR-11-LABEX-0043 » and ANR-10-IDEX-0001-02 PSL* and A*MIDEX project ANR-11-IDEX-0001-02 funded by the "Investissements d'Avenir" French government program. Grant from French Agency for Research on AIDS and Viral Hepatitis (ANRS) ECTZ88500 "SMARTHCV" also supported this project. The research is supported by the Ilídio Pinho foundation, Maratona da Saúde and FCT—Fundação para a Ciência e a Tecnologia—and Programa Operacional Competitividade e Internacionalização—Compete2020 (FEDER)—references PTDC/BIA-CEL/28791/2017 and POCI-01-0145-FEDER-028791, POCI-01-0145-FEDER-030882 and PTDC/BIA-MOL/30882/2017 and UIDB/04501/2020. We thank Lionel Spinelli and Thien-Phong Vu-Manh at CIML for bioinformatics and statistics support. We acknowledge financial support from no ANR-10-INBS-04-01 France Bio Imaging and the ImagImm CIML imaging core facility. The authors declare to have no competing interest.

### Author Contributions

A Mendes: conceptualization, formal analysis, validation, investigation, methodology, and writing—original draft, review, and editing.
JP Gigan: conceptualization, formal analysis, investigation, methodology, and writing—original draft, review, and editing.
C Rodriguez Rodrigues: conceptualization, formal analysis, investigation, and methodology.
SA Choteau: software, formal analysis, validation, investigation, and methodology.
D Sanseau: conceptualization, formal analysis, investigation, and methodology.
D Barros: conceptualization, formal analysis, investigation, and methodology.
C Almeida: conceptualization and investigation.
V Camosseto: conceptualization, formal analysis, investigation, and methodology.
L Chasson: formal analysis, investigation, and methodology.
AW Paton: resources.
JC Paton: resources.
RJ Argüello: conceptualization, investigation, and methodology.
A-M Lennon-Duménil: conceptualization and methodology.
E Gatti: formal analysis.
P Pierre: conceptualization, formal analysis, supervision, funding acquisition, validation, investigation, methodology, writing—original draft, and project administration.

### Conflict of Interest Statement

The authors declare that they have no conflict of interest.

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
