## [Reviewer comments · Life Science Alliance]

Life Science Alliance

Proteostasis in dendritic cells is controlled by the PERK signaling axis independently of ATF4

Andreia Mendes, Julien Gigan, Sébastien Choteau, Doriane Sanseau, Daniela Barros, Catarina Almeida, Voahirana Camosseto, Lionel Chasson, Adrienne Paton, James Paton, Rafael Argüello, Ana-Maria Lennon-Duméni, Evelina Gatti, and Philippe Pierre

DOI: [10.26508/lsa.202000865](https://doi.org/10.26508/lsa.202000865)

Corresponding author(s): *Dr. Philippe Pierre, (CIML-Centre d'Immunologie de Marseille Luminy)*

Review Timeline:	Submission Date:	2020-07-29
	Editorial Decision:	2020-08-03
	Revision Received:	2020-11-09
	Editorial Decision:	2020-12-07
	Revision Received:	2020-12-10
	Accepted:	2020-12-10

Scientific Editor: *Shachi Bhatt*

Transaction Report:

Please note that the manuscript was reviewed at Review Commons and these reports were taken into account in the decision-making process at Life Science alliance.

Review
COMMONS

1st Authors' Response to Reviewers

“Proteostasis in dendritic cells is controlled by the PERK signaling axis independently of ATF4” by Mendes et al.

We thank all the reviewers for their constructive comments on our work. We have tried to address as well as we can their experimental concerns and provide detailed answers to their remarks.

Reviewer #1 (Evidence, reproducibility and clarity (Required)):

****Summary:****

In their manuscript "Developmentally regulated PERK activity renders dendritic cells insensitive to subtilase cytotoxin-induced integrated stress response", Mendes et al. address the biochemical basis for the maintenance of high protein translation in response to triggers usually associated with the integrated stress response in dendritic cells. As the integrated stress response usually induces translational arrest, the DC's ability to overcome its induction is presumably vital for its function upon activation. The authors find that DCs display a very high level of PERK-mediated eIF2a phosphorylation at steady-state, which decreases upon LPS stimulation, correlating with the increased ability to synthesize protein upon activation. Despite this high basal eIF2a phosphorylation, DCs do not synthesize ATF4 and do not engage in the integrated stress response. The translational capacity of DCs in the steady state is controlled by GADD34. Finally, the authors provide evidence of an interplay between eIF2a phosphorylation, actin polymerization, and cytokine production and migration by the DC. To address their various questions, the authors perform numerous in vitro experiments, using an extensive range of inhibitors and activators in combination with transgenic models. In their conclusions, the authors mostly rely on western blotting and other protein biochemistry readouts. The authors however also use ex vivo single cell analysis by flow cytometry to validate their models, and turn to gene expression profiling, to strengthen their findings.

****Major comments:****

In this study, the authors provide a massive amount of data, which alleviates some of the shortcomings caused by marginal differences observed in some readouts. While I appreciate the plethora of data in the manuscript and agree with the necessity of the different experimental angles, it does make the manuscript very hard to follow. This is further enhanced by the fact that the field uses an enormous amount of abbreviations. The authors should check the manuscript carefully for possible options to streamline the narrative to facilitate the reading.

We have now rewrote the entire manuscript to answer this critics that was common to two of the reviewers.

The title of the manuscript on the pdf (but not on the user interface) include the claim that the developmentally regulated PERK activity is what renders the DCs insensitive to subtilase cytotoxin-induced integrated stress response. Where is the direct link between PERK and this resistance shown? Figure 7b seems to suggest that cDC2 maintain high levels of protein synthesis in the absence of PERK. Another point of unclarity is the meaning of "developmentally regulated", which seems solely based on the increasing expression over time in FLT3L cultures. What is the relevance of this point? If it is relevant enough to be listed in the title, it should be addressed more thoroughly along the developmental path of DCs rather than just in the final product of DC maturation. Both of these points may not require further experiments but rather clarification.

We have now changed the title to “Proteostasis in dendritic cells is controlled by the PERK signaling axis independently of ATF4”, which describes more accurately our observations. Figure 7B suggests that translation is poorly affected by SubAB when compared to thapsigargin, which occurs even upon PERK deletion. This was now clarify in the text (see Fig. 7D). Concerning the acquisition of eIF2a phosphorylation during DC development, we have shown that phosphorylation was strongly augmented upon appearance of the different DC subsets after several days of bone marrow progenitors differentiation *in vitro*. We agree with this reviewer, that the characterization of this phenomenon is probably insufficient to mention it in the title, despite the demonstration in Fig. 2 that protein synthesis acquisition is

clearly linked to DC differentiation, which could be considered as “developmentally regulated”, but we choose nevertheless to omit this part and change the title accordingly.

A major concern regarding the current manuscript is the stringency of data analysis. Many conclusions rely on the quantification of immunoblotting. It is unclear how this was done, and data is presented in the bar graphs. The figure legends state that experiments were done three times. Statistics or at least depiction of the variability between the experiments in the quantitative assessments would be very useful to better interpret the data. As an example, it is very hard to match the plots in Fig. 5A to the quantification. Time point 2 and 4 seem very different on the plot, but not in the bar graph. Does this reflect experimental variation?

We have chosen to display the best quality western blots available for Figure 5, together with the statistics related to the several experiments performed. We have now clarified how quantification was performed and added all the missing quantitative statistics by repeating experiments to decrease experimental variation, when necessary.

The authors employed T tests throughout the manuscript as statistical method, which is almost never appropriate, given the comparison of multiple groups per experiments, requiring multiple comparison testing. Running for reliable statistical tests and showing more rigorous quantification may require the repeat of some experiments in which differences were minor. The number of replicates should be stated in the figure legends throughout the manuscript.

The indication of using T-tests through-out the manuscript was inaccurate. Data were analyzed using the most appropriate statistical tests including Dunnett’s multiple comparison, Mann-Whitney and Wilcoxon tests. The number of replicates and type of statistical tests is now mentioned in all the legends.

****Specific comments:****

Figure 1:

(C) What does "relative MFI" refer to? The sentence starting with "DC subsets display higher levels...." should be removed as results should not be stated in the legend.

We show the quantitation of several flow analysis equivalent to the one presented next to this quantification in Figure 1C, the relative MFI is calculated with reference to B cells as 100%. The sentence was removed from the legend.

(F) The authors claim higher protein synthesis in all resting DCs compared to activated T cells. This does not seem to be reflected the figure.

This was an overstatement that escape our proof reading, the sentence is now corrected for accuracy.

Last sentence of text relates to the comment on the title: The reasoning for bringing this up is unclear to me given that all DCs arise from BM progenitors and eIF2a expression phosphorylation was not assessed on a per cell basis along a developmental path (which very well might be impossible to do and is certainly not requested).

Indeed, detection of eIF2 α phosphorylation by flow cytometry is technically challenging, and insufficiently sensitive to complete an analysis of rare DC progenitors as underlined by this reviewer, we therefore changed the text accordingly.

Figure 2:

(C) The authors point-out increased eIF2B over eIF2a ratio, but do not take into account the massive accumulation of p-eIF2a. Is that a fair comparison? Along those lines, the ratio of the mRNA level does not seem to change (B).

We have now added novel qPCR data for the mRNA expression of additional eIF2B subunits as well as of eIF2A to found a similar increase during DC differentiation. We also quantified

the protein ratios in Fig. 1C using P-eIF2 α levels rather than eIF2 α , to reach the same conclusions.

Figure legend:

Cells were incubated for...neighbor embedding was applied... in black are gated...cDC2 in blue...stained in green...eIL2B/eIF2S1 is shown...(typos and small mistakes)

The sentence "In parallel to..." should be deleted as it is stating results.

This was corrected in the new figure legend.

Text file: The switching between protein and gene name for eIF2 makes it difficult to follow without introduction

We have simplified the wording to avoid these difficulties.

Figure 3:

Legend:

The sentence "Phosphorylation of both..." should be deleted as it is stating results.

Are cDC2s indeed gated as CD24+? Or should this say -?

The part "showing that the speed of elongation..." should be deleted as well

This was corrected in the new figure legend. In fact the cDC2 are CD24 low, but to avoid confusion we indicated their CD11b positivity rather than mentioning CD24. The Figure 3 has now been reorganized to answer the suggestions of reviewers 1 and 3.

Figure text:

Starting the paragraph with the sentence "Complementary to p-eIF2a..." might facilitate reading as the start is abrupt without an introduction of the thought.

Last paragraph: ", we applied to cDC2," is situated wrongly in the sentence. Is that even true? The figure suggests this was done on all DCs.

This was corrected in the text and adapted for a differently organized Figure 3 with a new figure legend. We removed Figure 1D (immunoblot of bulk cells) to focus on the flow analysis of cDC2, which is more informative and for which we provide the slope values.

Figure 4:

In (A), the dephosphorylation in response to LPS is minor when compared to what was shown in Figure 3. This makes it hard to interpret the depicted data.

We have now combined old Fig. 4 and Fig. 5 in a new Figure 4. We realized that most of the data presented on old Figure 4 were not necessary to support the main findings of the paper and could distract the attention of the readership. We therefore decided to remove most of them, in the spirit of simplifying our message and improving the readability of the text. We only kept the old panel 4A (blot) and 4B (PCR GADD34), that are now new panels 4C and 4D describing that GADD34 expression requires TBK1 activity.

Concerning new Fig 4C (old 4A), the blot contrast can be deceptive and the quantitation indicates at least a 50% eIF2 α phosphorylation is lost in the first 4 hours of the treatment. Importantly, given that LPS stimulates mostly cDC2 in the bulk culture, the activation of the other DC populations occurs indirectly, mostly in response to type-I IFN secretion, thus some variations are therefore always observed in the bulk rate of eIF2 α dephosphorylation, due both to the proportion of cDC2 present in the culture and the amount of IFN- β produced. Irrespective of these variations eIF2 α dephosphorylation is always observed upon activation in all DC subsets.

In (C), is there a particular reason why the time points checked were later than those for GADD34?

The induction kinetics of GADD34 mRNA is more rapid than that of ATF4 or CHOP, which are poorly induced in DCs, so the time points were adapted to this situation, we nevertheless removed the data concerning ATF4 and CHOP, which were not necessary with the aim of improving the coherence and focus of the manuscript.

Figure text:

"p-eIF2a levels are clearly augmented even in the absence of LPS..." - they seem augmented only in the absence of LPS.

The part referring to cytokine expression is unclear. Can the authors embed their thought about elevated MHCII levels because of changed cytokines better?

New immunoblots were performed to improve the quantification and support our claims in now new Figure 4C. Again, we removed the MHC II expression from old Fig. 4, given the impossibility to propose a convincing and rational explanation for this particular effect of TBKin. The text was simplified and adapted to the new Figure 4 with the aim of refocusing the manuscript.

Figure 5:

Can the authors explain how normalization is done in the quantifications?

The normalization was done with respect to the level of non-stimulated WT DCs ratios.

(B) Quantification of the slope could potentially strengthen the claim.

The slopes on the SunRISE plots have now been added and give information on the rate of polysomes processing along the mRNA.

Text:

"...nor their respective impact on translation" (bottom page 14). Was this sufficiently assessed here to allow that claim?

We agree with this reviewer and this sentence was eliminated during the text editing.

Figure 6:

(A) Why do dPERK cells still show 5 times more protein than some of the controls?

As in every Cre-deletion model, gene inactivation is not always perfect *in vivo*. In particular, in the CD11c-CRE mouse, various level and kinetics of CD11c-Cre expression has been observed according to the cell subset studied. This variability has obvious consequences on the efficacy of floxed sequences excision by the Cre recombinase. It is therefore normal to find remaining expression of PERK in a minority of CD11c+ splenocytes, that have low expression or have acquired CD11c expression late during their differentiation. This population obtained by magnetic sorting is also contaminated by a small proportion of PERK+ CD11c- cells. Importantly, the excision of the PERK gene is extremely efficient *in vitro* as observed in the Flt3-L-BMDC.

Legend:

The sentence "Not-activated PERKdK..." should be deleted as it states results.

This was deleted from the legend.

Figure 7,

text:

"A modest PERK-dependent induction of p-eIF2a was observed in WT cells" - this is difficult to make out from the data presented in the figure.

The quantification of Fig 7A (now new 6A) shows a modest increase in p-eIF2 α after 2h of SubAB treatment but not much else, our comment is therefore accurate.

"BiP" deserved a more elaborate introduction.

A sentence describing BiP function was added in the chapter.

Figure 8,

legend:

The sentence "ISIRB treatment was efficient..." should be deleted as it is stating results.

The sentence was deleted.

Figure 9:

In (A), the time point at baseline is poor for Jasplakinolide, so interpretation is difficult. However, it still looks like phosphorylation is decreasing rather than increasing, and with that is matching, rather than opposing the effect of Latrunculin.

We have included the quantification of the blots to show that jasplakinolide increases eIF2 α phosphorylation and Latrunculin A decreases it.

Text:

The authors claim that there is a synergistic effect between LPS and Lat A treatment, but the differences seem very subtle. This might thus be an overinterpretation of the data.

We agree with this reviewer and have corrected the text to match the data.

Suggestion for sentence fix: "These results further suggest...could be key regulating factors for the homeostasis and translation of specific mRNA encoding for proteins generally secreted in a polarized fashion, such as type I IFNs, or associated with cell migration."

The last reference to Fig. 9G should read 9H.

We thank reviewer 1 for his/her suggestion and this was corrected.

Discussion:

"global or local translation" - It is somewhat unclear what that refers to.

The sentence was removed.

Reviewer #1 (Significance (Required)):

This is a very detailed analysis on the integrated response of dendritic cells to stimulation. Dendritic cells have the unique capability of inducing naive T cells, a function that requires finely tuned integrated responses to environment for faithful T cell instruction and the ability to migrate. The current knowledge regarding how dendritic cells can support these various important functions from a biochemical perspective is rather incomplete. The presented data is novel and provides some answers while raising many additional questions, opening venues for future investigations. The topic is complex and requires a high level of specialization, but it is highly relevant regarding its possible applications.

My area of expertise is DC subset biology and function, but it does not include stress responses or intracellular signalling and I therefore feel not qualified to judge whether all conclusions made are appropriate. I mentioned my concerns wherever in doubt, hoping that this input will help the authors to clarify some of those aspects of their study where I could not follow the reasoning. I therefore deliberately formulated many comments as questions.

We thank this reviewer for his/her encouraging comments about the relevance of our work, and have found his/her comments to be fair and completely justified. We did our best to answer these critics by including many modifications in the figures or the text and by performing novel experiments.

REFEREES CROSS COMMENTING

I agree with the other reviewer's comments. We all pointed out that the data is interesting, but that the analysis is often not stringent enough and that the manuscript is in part hard to follow. The suggested additional experiments are clearly of value and I think the authors will benefit from this feedback, even if suggesting additional experiments may not be in the spirit of ReviewCommons. All in all, I do not think that there is a need for extensive cross consultation in this case - all three reviews are thorough and sensible.

We agree with this statement and have performed antigen presentation assays to complete this novel version of the manuscript and finalize our study. We are however encountering breeding difficulties with several strains of mice due to the arrest of our activities during the COVID-19 crisis and still can only at the present time perform these experiments using pharmacological inhibitors as surrogates for PERK genetic inactivation. Our data indicates that PERK-inhibited DCs do not display any impairment in MHC II restricted presentation of antigenic peptides or soluble antigens to T cells (New Supplementary Figure 9).

Reviewer #2 (Evidence, reproducibility and clarity (Required)):

Mendes et al show that differentiated dendritic cells (DCs) display unusually high eIF2 α phosphorylation, but normal protein biosynthesis and cytokine production. They also show that high eIF2 α phosphorylation is caused by a developmentally regulated activation of PERK. Moreover, they show that activation of the PERK-ISR prevents translation arrest and ATF4 expression during ER-stress induction by subtilase cytotoxin or upon DC stimulation with bacterial lipopolysaccharides, and influences the actin cytoskeleton dynamics in DCs. The topic is important. However, there are a number of major concerns.

We thank this reviewer for his/her encouraging comments about the importance of our work, and have tried to address all his comments.

1 . It is an interesting phenomenon that differentiated DCs display unusually high eIF2 α phosphorylation. Although the authors present some data showing the potential physiological function of high eIF2 α phosphorylation in DCs, including regulating cytokine production and actin cytoskeleton dynamics, its physiological significance is unclear. The manuscript would be improved considerably by uncovering the physiological significance of high eIF2 α phosphorylation in DCs, particularly in antigen presentation process.

Our analysis was very thorough and the results point towards a general control of DC homeostasis by PERK and GADD34 with no major immunological phenotypes *in vitro* revealed by their inactivation. However this inactivation causes many small defects, likely to be all relevant for the adaptation of DCs to highly variable environments and specific physiological conditions difficult to reveal in our experimental setting. Notably, the surprising observation that PERK deficiency impacts both IFN- β secretion and DC migration is a completely novel finding that is going to re-orientate the focus of our research towards understanding better the link existing between actin dynamics and protein synthesis regulation. In addition, the different reviewers failed to underline, that we were unable to confirm in primary DCs, that the ISR is necessary to trigger cytokines expression and secretion as it was recently proposed for macrophages in two recent major publications. Although these data do not clarify the physiological importance of the ISR for DC function, they reveal that together with their inability to mount an ISR upon SubAB toxin exposure, DCs display particular features allowing them to function in adverse biochemical situation or environment. We nevertheless agree with this reviewer on the importance of exploring the consequences of PERK inactivation on antigen presentation by DCs and have performed accordingly a series of experiments to finalize our work. Our results indicate that acute PERK inhibition has no effect on MHC II-restricted antigen presentation and T cell activation *in vitro* (New Supplementary Figure 9).

-

2 . There is no *in vivo* data in this manuscript, except immunostaining images in figure 1. The authors generate DC-conditional PERK and GADD34 knockout mice, it is important to use these mouse models to generate some *in vivo* data to back up the *in vitro* findings.

We respectfully consider that initiating *in vivo* experiments at this stage is not going to change our conclusions and will require too much time and animal experimentations to be performed in our current working conditions, that are still deeply impacted by the COVID19 crisis. We consider that the breadth of our work is sufficient to have the manuscript stands on its own, although in the future, our results will be used to generate different hypothesis that will be tested *in vivo*.

3 . Figure 8 and 9 show that the effects of the PERK-ISR on innate receptor signaling and actin cytoskeleton dynamics in DCs are modest. The magnitude of changes is very modest, less than 50%.

Although reviewer 2's comment is general and it is difficult to answer specifically, we would not consider as modest, a loss of 50% in protein synthesis, like the one induced by Jasplakinolide (Fig 10B). In our view, the loss of one neo-synthesized protein out of two is likely to have major consequences on cellular proteostasis and cellular functions. As for Fig. 8 (now new Fig . 7), our up-dated results show that IFN- β is inferior by more than 50% in PERK Δ K cells or upon PERK inhibition compared to control. The differences observed for some other cytokines upon pharmacological inhibition of PERK are also significant (new Fig 8). Importantly, our observations indicate that interfering with PERK expression or the ISR has highly variable consequences for cytokine expression which are dependent on the type of the cytokine, whether the ISR is inhibited pharmacologically or genetically, and on the cell type studied. Importantly, our observations in DCs contradict the results published by Abdel-Nour et al. (Science, 2019) and Chiritoiu et al. (Developmental Cell, 2019) and underline the complexity of generalizing our observations to all immune cells subsets.

4 . The authors should provide quantification data for western blot, including SD and p value.

We have now provided quantification and statistics, where missing.

5 . The authors should provide the matched data sets in figures. For example, flow vs flow, western blot vs western blot. Many data sets in manuscript are hard to follow, because the authors frequently show flow vs western blot.

We have tried to follow this advises, although the presentation of both Immunoblots and flow data is a good experimental practice to evaluate variations and allow more accurate quantification. With this aim, we have replaced the immunoblots in Fig 3A and 3B and 3D by equivalent flow plots and the corresponding error bars.

Reviewer #2 (Significance (Required)):

Although the manuscript describes an interesting phenomenon, namely unusually high eIF2 α phosphorylation in DCs, the authors fall short of uncovering its physiological function. There is no effort to determine whether high eIF2 α phosphorylation is required for antigen presentation or other important function of DCs.

We have provided many important information on the regulation and the consequences of the ISR in relevant primary dendritic cells. Furthermore, we have reveal a potential novel function of PERK in regulating the migration of these cells and the existence of a cross-talk between the ISR and actin organization. We have also provided a novel analysis of the antigen presentation function of DC exposed to PERK inhibition, however without showing any significant alteration of this function of soluble exogenous antigens, despite a clear impact on IFN- β expression.

Reviewer #3 (Evidence, reproducibility and clarity (Required)):

Mendes et al. have investigated the importance of pathways related to the ISR in dendritic cells. They found that DCs display unusually high eIF2 α phosphorylation due to developmentally regulated activation of PERK. Despite that, differentiated DCs do not display signs of a chronic ISR and have an active protein synthesis. Rather, eIF2 α appears to be important in adapting protein homeostasis to the variations imposed on DCs by various contexts. This biochemical specificity prevents translation arrest and expression of the transcription factor ATF4 during ER-stress induction by subtilase cytotoxin or upon DC stimulation with bacterial lipopolysaccharides. Although this is a carefully performed and comprehensive study, the overall aim and message is not clear enough and needs some rewriting (including the abstract). If the message is that these features make DCs resistant to ER stress, increase survival, migration capacity etc, that has to be said. In addition, the following issues need to be addressed.

We thank this reviewer for his/her encouraging comments about the importance of our work, and have tried to address all his/her concerns, starting with rewriting entirely the manuscript.

****Specific comments****

1 . A more comprehensive analysis of whether cytokines are affected needs to be performed. Analyzing TNF, IL-10 and IL-12 would add to the study.

We fully agree with reviewer 3 and added measurement in a new figures 7 and 8 of many different cytokines in conditions of pharmacological inhibition of PERK with the specific inhibitor GSK2656157 or upon ISRIB exposure. We have added some measurement for the PERK-deficient cells in new figure 8. Unfortunately, we have encountered problem to maintain our mouse breeding capability during the COVID 19-confinement and are unable to perform more of these type of experiments for a relatively long period, since we have to re-derived several strains from frozen sperm stocks.

2 . Also, why haven't the authors looked at cytokine protein levels? This is important and need to be addressed.

We agree with this reviewer and all the cytokines expression measurement for IFN β , IL-6 and IL-1b were already performed for mRNA and protein in old Fig. 8 and 9 of the manuscript, where protein expression measurement were required. In this new version of the manuscript, we have also included TNF and IL-10 secretion levels in new Figure 8, while mRNAs expression is documented in a new Sup Fig 7.

3 . If Fig. 1C the flow plots of the p-eIF2 α staining patterns need to be shown, at least in the Supplemental information.

We have now reorganized Figure 1C and added a flow profiles for one representative experiment, in addition to the quantitation plots that include the results of several experiments.

4 . In Fig 2C, why aren't error bars shown in the Quantification of the ratio eIF2B/eIF2 α graph shown? Although the WB has to be a representative of n=3 independent experiments, in the quantification plot all 3 experiments can be taken into account.

Error bars have been added for the quantification, but we now provide the ratio for eIF2B/P-eIF2 α since it is more representative than eIF2 α .

5 . How does the fluorescent pattern in flow cytometry detection of puromycin (SunRISE) look like? Again, that should be included in the Supplemental Information.

We have now added the slope values for the SunRISE data to improve the interpretation of the data and provide the flow profiles in a new Figure 3C and 3D.

6 . Is antigen-presenting capacity affected? It would be good to address that e.g. in an antigen-specific or MLR-type of assay.

As requested by several reviewers, we have now performed antigen presentation assays upon PERK inhibition. Our results indicate that MHC II antigen presentation of soluble antigens is not affected by GSK2656157.

Reviewer #3 (Significance (Required)):

The study is of interest and potential significance, the experiments are well controlled and comprehensive. The only problem is that it does not convey a clear message for the aim and main conclusions. With a bit of rewriting that can be very much improved.

We thank this reviewer for his/her supportive comments and have rewrote the manuscript entirely to respond to his/her last comments.

August 3, 2020

Re: Life Science Alliance manuscript #LSA-2020-00865

Dr. Philippe Pierre
CIML-Centre d'Immunologie de Marseille Luminy
CIML-Centre d'Immunologie de Marseille Luminy
163, avenue de Luminy
Parc Scientifique et Technologique de Luminy 163, avenue de Luminy case 906
Marseille 13288
France

Dear Dr. Pierre,

Thank you for submitting your manuscript entitled "Proteostasis in dendritic cells is controlled by the PERK signaling axis independently of ATF4" to Life Science Alliance. We have now carefully read your study, the referees' reports from Review Commons, as well as your point-by-point rebuttal letter. We find that your plan to address the referees' concerns appears to be reasonable.

Given the overall interest of your study, we would thus invite you to revise the manuscript as indicated in the reviews. While we do not request you to perform in vivo experiments, addressing the physiological significance of high eIF2 α phosphorylation in dendritic cells during antigen presentation would be essential for publication here. Your manuscript will be re-assessed by the original referees and we will need strong support from them; no new issues will be raised.

In our view these revisions should typically be achievable in around 3 months. However, we are aware that many laboratories cannot function fully during the current COVID-19/SARS-CoV-2 pandemic and therefore encourage you to take the time necessary to revise the manuscript to the extent requested above. We will extend our 'scoping protection policy' to the full revision period required. If you do see another paper with related content published elsewhere, nonetheless contact me immediately so that we can discuss the best way to proceed.

Thank you for this interesting contribution to Life Science Alliance. We are looking forward to receiving your revised manuscript.

Sincerely,

Reilly Lorenz
Editorial Office Life Science Alliance
Meyerhofstr. 1
69117 Heidelberg, Germany
t +49 6221 8891 414
e contact@life-science-alliance.org
www.life-science-alliance.org

B. MANUSCRIPT ORGANIZATION AND FORMATTING:

December 7, 2020

RE: Life Science Alliance Manuscript #LSA-2020-00865R

Author information redacted

Dear Dr. Pierre,

Thank you for submitting your revised manuscript entitled "Proteostasis in dendritic cells is controlled by the PERK signaling axis independently of ATF4". We would be happy to publish your paper in Life Science Alliance pending final revisions necessary to meet our formatting guidelines.

Along with the points listed below, please also attend to the following,

- please make sure that the author order in your manuscript and in the system match
- please consult our Manuscript Preparation Guidelines <https://www.life-science-alliance.org/manuscript-prep> and put your manuscript sections in the correct order
- please add ORCID ID for secondary corresponding author-they should have received instructions on how to do so
- please add a separate conflict of interest statement to your main manuscript text
- please upload both your main and supplementary figures as single files
- please upload your tables as editable doc or excel files
- please make sure your final manuscript text is uploaded in editable doc format
- please use the [10 author names, et al.] format in your references (i.e. limit the author names to the first 10)
- please add a callout for Figure 9H in your main manuscript text
- please revise the legend for figure 5 so that the panels are introduced in order
- please revise the inset position in Figure 1B so that they match the zoomed in parts

A. FINAL FILES:

B. MANUSCRIPT ORGANIZATION AND FORMATTING:

Sincerely,

Shachi Bhatt, Ph.D.
Executive Editor
Life Science Alliance
<https://www.lsjournal.org/>
Tweet @SciBhatt @LSAJournal

Reviewer #1 (Comments to the Authors (Required)):

The authors have adequately addressed all my concerns to the extent possible. There is no further comments.

Reviewer #2 (Comments to the Authors (Required)):

The manuscript "Proteostasis in dendritic cells is controlled by the PERK signaling axis independently of ATF4", from Dr. Phillip Pierre and colleagues was previously assessed via Review Commons by three independent reviewers.

Following their valuable and constructive comments and advises, the manuscript has been extensively restructured and re-formulated. New data, new statistical analysis, and additional quantifications as well as figures have been added to solidify their data and conclusions.

Due to current COVID-19 pandemic some of the suggested experiments can't be performed in due time therefore I fully support the publication of this manuscript in LSA as it currently is.

December 10, 2020

RE: Life Science Alliance Manuscript #LSA-2020-00865RR

Author information redacted

Dear Dr. Pierre,

Thank you for submitting your Research Article entitled "Proteostasis in dendritic cells is controlled by the PERK signaling axis independently of ATF4". It is a pleasure to let you know that your manuscript is now accepted for publication in Life Science Alliance. Congratulations on this interesting work.

DISTRIBUTION OF MATERIALS:

Again, congratulations on a very nice paper. I hope you found the review process to be constructive and are pleased with how the manuscript was handled editorially. We look forward to future exciting submissions from your lab.

Sincerely,

Shachi Bhatt, Ph.D.
Executive Editor
Life Science Alliance

<https://www.lsjournal.org/>
Tweet @SciBhatt @LSAJournal